# Longitudinal imaging of HIV-1 spread in humanized mice with parallel 3D immunofluorescence and electron tomography

Collin Kieffer, Mark S Ladinsky, Allen Ninh[†], Rachel P Galimidi, Pamela J Bjorkman*

Division of Biology and Biological Engineering, California Institute of Technology, Pasadena, United States

**Abstract** Dissemination of HIV-1 throughout lymphoid tissues leads to systemic virus spread following infection. We combined tissue clearing, 3D-immunofluorescence, and electron tomography (ET) to longitudinally assess early HIV-1 spread in lymphoid tissues in humanized mice. Immunofluorescence revealed peak infection density in gut at 10–12 days post-infection when blood viral loads were low. Human CD4+ T-cells and HIV-1–infected cells localized predominantly to crypts and the lower third of intestinal villi. Free virions and infected cells were not readily detectable by ET at 5-days post-infection, whereas HIV-1–infected cells surrounded by pools of free virions were present in ~10% of intestinal crypts by 10–12 days. ET of spleen revealed thousands of virions released by individual cells and discreet cytoplasmic densities near sites of prolific virus production. These studies highlight the importance of multiscale imaging of HIV-1–infected tissues and are adaptable to other animal models and human patient samples.

*For correspondence: bjorkman@caltech.edu

Present address: [†]Icahn School of Medicine at Mount Sinai, New York, United States

## Introduction

HIV-1 presents an important global health issue with >36 million people infected worldwide and 2.1 million newly acquired infections in 2015. The majority of new HIV-1 infections result from heterosexual transmission, with women accounting for a majority of all new infections worldwide (**UNAIDS, 2015**). During heterosexual transmission of HIV-1, the virus crosses the genital epithelium and infects a target cell, which results in localized infection, establishment of a latently-infected virus reservoir, expansion of a local founder population, and systemic virus spread to distant tissues via lymph and blood (**Fackler et al., 2014**). With blood containing only 1–2% of HIV-1 target cells in the body, the majority of HIV-1 target cells in the human body reside within tissues (**Guy-Grand and Vassalli, 1993**; **Kilby, 2001**). Gut-associated lymphoid tissue (GALT), lymph nodes, and spleen contain a large number HIV-1 target cells that support localized virus transmission and remain a key source of virus replication during systemic spread and long term disease progression (**Haase, 1999**; **Reinhart et al., 1997**; **Grossman et al., 1998**), yet the biological details of interactions between the virus and immune cells within tissues remain poorly characterized at single cell and subcellular resolution.

HIV-1–infected animal models offer an opportunity to investigate virus spread within tissues using immunofluorescence (IF) and electron microscopy (EM) imaging. Model systems of HIV-1 infection include both non-human primates (NHP) and mice, with mouse models being more practical for longitudinal studies of infection in which relatively large numbers of animals are infected. Mice with humanized immune systems (hu-mice) allow the rapid generation of human target cells infected with

wildtype virus in a living organism with a functioning immune system and recapitulate important aspects of HIV-1 pathology including mucosal infection, systemic virus spread, latency, and response to therapeutic intervention (*Hatziioannou and Evans, 2012*; *Marsden and Zack, 2015*).

Optical clearing techniques, originally developed >100 years ago, enabled imaging of large volumes of tissue, which led to an enhanced understanding of human anatomy, but the original clearing methods are not compatible with modern fluorescence microscopy due to tissue destruction and quenching of the fluorescent signal (*Richardson et al., 2015*; *Treweek and Gradinaru, 2016*; *Tainaka et al., 2016*). However, recent advances in optical clearing of intact tissues combined with 3D fluorescence microscopy enable imaging of tissues at single cell resolution (*Richardson et al., 2015*; *Tainaka et al., 2016*). The basic principles of tissue clearing involve removing opaque biomolecules that reduce light penetration into samples and matching the refractive index of optics, solutions, and tissues to minimize optical distortions during imaging. CLARITY/PACT and CUBIC tissue-clearing techniques were recently applied to render entire rodents optically clear while maintaining tissue integrity at the single cell level (*Chung et al., 2013*; *Susaki et al., 2014*; *Tainaka et al., 2014*; *Treweek et al., 2015*; *Yang et al., 2014*). When combined with IF and fluorescence microscopy, these techniques allow imaging of large volumes ($mm^3$–$cm^3$) of intact tissues at single cell resolution, delineation of the connectivity of biological structures, and discrimination of individual cell types within complex tissue environments. Tissue clearing techniques were recently applied to image brain, nervous system, and internal organs in rodents, the spatial heterogeneity of mouse tumor models, developmental processes in zebrafish, and post-mortem pathology from preserved human samples (*Tainaka et al., 2016*; *Chung et al., 2013*; *Susaki et al., 2014*; *Tainaka et al., 2014*; *Tomer et al., 2014*; *Guldner et al., 2016*; *Tomer et al., 2015*), but have not previously been used to image lymphoid tissues from HIV-1–infected animals.

EM was originally utilized to identify HIV-1 as the causative agent of AIDS (*Barré-Sinoussi et al., 1983*; *Gallo et al., 1983*) and has subsequently been applied to examine aspects of HIV-1 pathology at ultrastructural resolution (*Orenstein, 2007*). ET can provide 3D information at subcellular resolution from frozen hydrated or from well-preserved, fixed biological samples (*McIntosh et al., 2005*). ET of purified HIV-1 and HIV-1–infected cultured cells have contributed to our understanding of the HIV-1 life-cycle through investigations of the structures of mature and immature virions (*Benjamin et al., 2005*; *Briggs et al., 2009*; *Wright et al., 2007*), envelope spike densities and conformations on virions (*Zhu et al., 2006*; *Liu et al., 2008*), virus biogenesis (*Carlson et al., 2010*, *2008*), cell entry, and cell-to-cell transmission (*Earl et al., 2013*). Although informative, these studies provided no information concerning HIV-1 in the context of an infection within a host organism. The in situ environment involves multiple cell types within tissues and organs, which may alter virus behavior and fine-structural associations compared to infection within a cultured cell environment. In a previous study, we combined optimized tissue preservation methods with ET to image HIV-1 in GALT derived from chronically infected hu-mice (*Ladinsky et al., 2014*). This approach allowed us to distinguish and characterize mature, immature, and budding virions, localize infected cells and intercellular pools of free virions within colon crypts, and identify distinct components of virus budding machinery within infected GALT tissues (*Ladinsky et al., 2014*). These studies shed light on HIV-1 interactions during later phases of infection, establishing the groundwork for investigations of HIV-1 during critical early virus spread in lymphoid tissues.

Here we address early HIV-1 spread by combining 3D-IF microscopy of optically-cleared tissues with ET of optimally-preserved tissues. This parallel approach allowed us to image systemic virus spread in HIV-1–infected hu-mouse lymphoid tissues at different volumes, resolutions, and times after infection. These analyses provided a longitudinal and spatial comparison of the location, type, and density of infected cells, routes of virus spread, and virus levels in lymphoid tissues. Infectivity in GALT and spleen preceded peak blood viremia and showed a rapid and exponential spread of virus in these tissues early after infection. Blood viral load peaked over a week later, when infectivity in several lymphoid tissues was beginning to diminish, emphasizing the discrepancy between blood and tissue infectivity. During times of peak infectivity, large numbers of free virions were identified in spleen, with individual infected cells being surrounded by >1000 virions within thin volumes of tissue. These studies provide multiscale information about the early systemic spread of virus within lymphoid tissues at both cellular and subcellular resolution, and these methods are readily adaptable to in situ analysis of HIV-1–infection in other hu-mouse and NHP models, and samples from human patients.

## Results

### Validation of PBMC-NSG hu-mouse model for imaging studies

Perhaps the most accurate hu-mouse model, bone marrow/liver/thymus (BLT) mice, are generated by transferring CD34+ human stem cells together with human fetal thymic and liver tissues during individual surgeries into immunocompromised mice (*Melkus et al., 2006*; *Lan et al., 2006*). While BLT mice are optimal for studies of mucosal transmission of HIV-1 (*Sun et al., 2007*; *Denton et al., 2008*), they are not cost effective for longitudinal studies of HIV-1 infection involving large cohorts of infected animals. We therefore used a technically simpler and less expensive hu-mouse model constructed by injection of primary human peripheral blood mononuclear cells/lymphocytes (PBMC/PBL) into NOD.Cg-*Prkd*scid*Il2rg*tm1Wjl/SzJ (NSG) mice (*Ishikawa et al., 2005*; *Shultz et al., 2005*; ; *King et al., 2008*), which are immunocompromised animals lacking mature natural killer cells, B-cells and T-cells. PBMC-NSG mice rapidly reconstitute human CD4+ and CD8+ T-cells in addition to human B-cells and disseminate human lymphoid cells to lymph nodes, spleen, bone marrow, genital mucosa, and intestine (*Hatziioannou and Evans, 2012*; *Marsden and Zack, 2015*; *Ishikawa et al., 2005*; *King et al., 2008*; *Mosier et al., 1988*). It is important to note that the PBMC-NSG hu-mouse model almost exclusively harbors activated human T cells that can react with host antigens and cause animals to develop graph versus host disease (*Marsden and Zack, 2015*). The lack of other cell lineages combined with raised levels of T cell activation in this model may enhance the dynamics of HIV spread and pathogenesis compared to other hu-mouse models that contain a smaller compartment of activated T cells at the time of infection and a more diverse set of immune cell lineages. Nonetheless, the robust reconstitution of activated human T-cells in the PBMC-NSG hu-mouse models is an important component for HIV-1 studies in hu-mice since (i) CD4+ T-cells make up the majority of all HIV-1 target cells in humans (*Haase, 1999*), (ii) activated T cells constitute a sizeable proportion of all CD4+ T cells in humans (~10%), (iii) activated T cells rise to ~20–40% during the course of HIV-1 infection (*Mahalingam et al., 1993*), and (iv) activated T cells localize to important tissue sites of virus pathology including spleen, lymph nodes, and GALT (*Saksena et al., 2010*; *Cerf-Bensussan and Guy-Grand, 1991*). Importantly, HIV-1 infection of PBMC-NSG mice mimics early stages of infection in humans, in that the virus replicates exponentially, resulting in large scale CD4+ T-cell depletion within a few weeks (*Marsden and Zack, 2015*; *Kim et al., 2016*; *Kumar et al., 2008*), and this model was previously used to study the effectiveness of anti-HIV-1 therapies in reducing systemic virus spread after intravenous transmission (*Kim et al., 2016*; *Kumar et al., 2008*; *Balazs et al., 2012*, *2014*; *Hur et al., 2012*). The lower cost, simplicity, rapid generation of HIV-1–infected animal cohorts, and effective reconstitution of systemic virus spread make the PBMC-NSG hu-mouse model advantageous for longitudinal studies investigating early T cell-mediated pathogenic events in GALT, spleen, and peripheral blood.

HIV-1-infected PBMC-NSG mice normally show robust blood p24 levels ~2–4 weeks post-infection (PI), followed by a sharp decrease of viral load (*Kim et al., 2016*; *Kumar et al., 2008*; *Balazs et al., 2012*, *2014*). By contrast, blood p24 levels in HIV-1 infected BLT mice regularly peak at ~4–6 weeks PI, followed by a slower reduction in blood viral load (*Sun et al., 2007*; *Brainard et al., 2009*; *Denton et al., 2008*). To compare with our previous imaging studies of GALT from long-term (10 to 20 weeks PI) infected BLT mice (*Ladinsky et al., 2014*), we chose three weeks PI for analysis and imaging in PBMC-NSG mice to ensure that human cells and virus would still be present in tissues. Female NSG mice were reconstituted with human PBMCs and three PBMC-NSG mice were infected with R5-tropic HIV-1 (pNL4-3envYU2). HIV-1 p24 levels in blood showed mean levels of 2.50 ± 0.78 ng/mL at 12 days PI and 12.68 ± 8.05 ng/mL at 21 days PI, demonstrating productive infection in all three animals (*Figure 1A*). An infected animal was necropsied at 21 days PI and adjacent GALT samples from the colon were fixed with aldehydes for parallel IF and ET analysis. IF of colon showed the presence of human CD3+ cells and HIV-1 p24 in infected animals three weeks PI in colon crypts and within the crypt epithelium (*Figure 1B*). ET corroborated the IF results, showing the presence of infected cells indicated by virus budding profiles projecting from the plasma membrane and free virions in similar tissue locations (*Figure 1C,D*). The localization and levels of T-cells and HIV-1 within colon crypts were consistent with our previous studies in HIV-1–infected BLT animals (*Ladinsky et al., 2014*), establishing the validity of the PBMC-NSG model for conducting longitudinal imaging studies of HIV-1 infection and spread in lymphoid tissues.

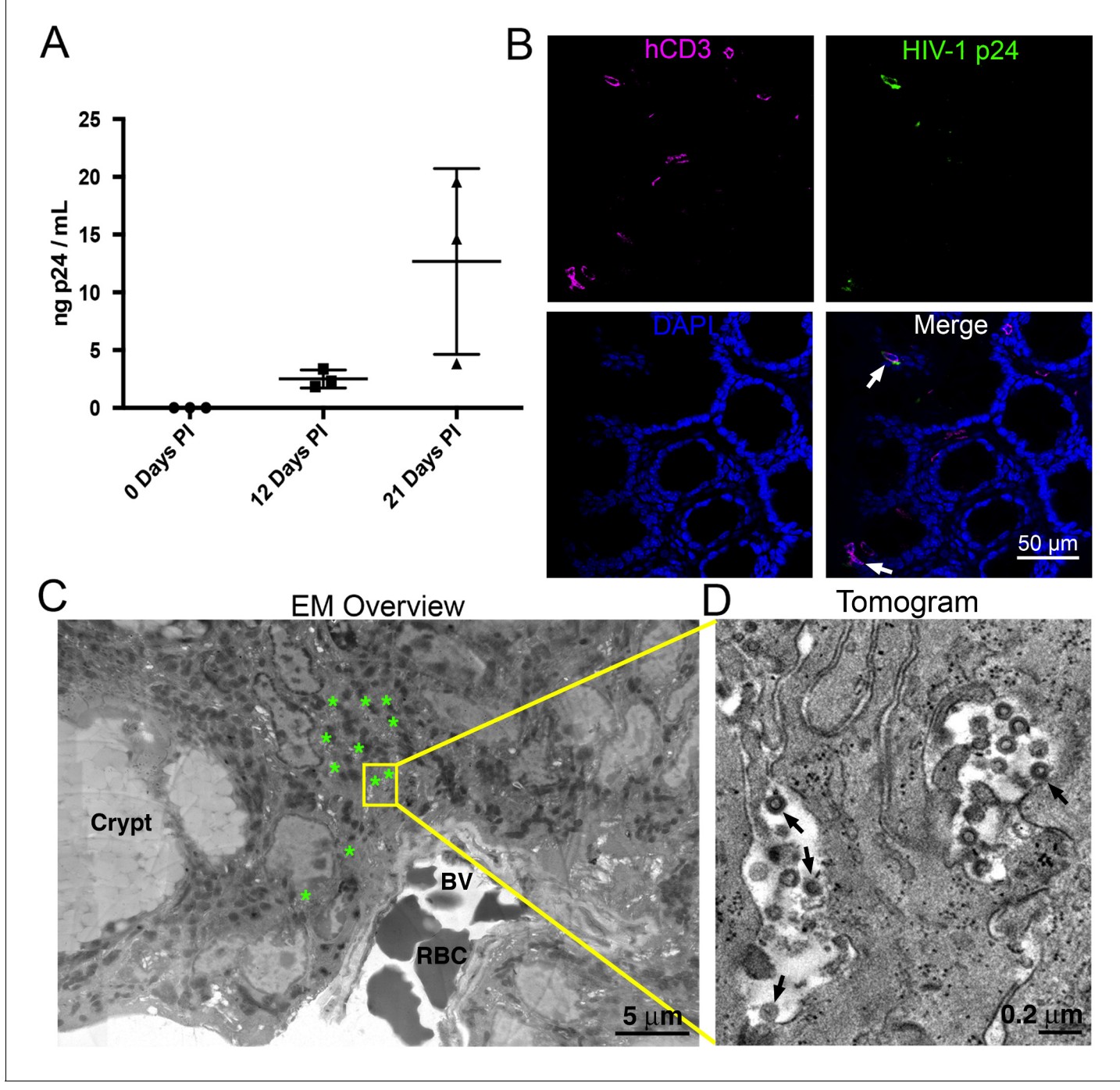

**Figure 1.** HIV-1 infection and Imaging of hu-mice. (**A**) Mean blood p24 levels from three HIV-1–infected PBMC-NSG mice assessed at 0, 12, and 21 days PI. Error bars represent standard deviation. (**B**) IF of GALT from HIV-infected animal with highest blood p24 level from panel A at 21 days PI. Samples were stained for human CD3 (magenta), HIV-1 p24 (green), and nuclei (blue). White arrows indicate human CD3+/p24+ cells. (**C**) Montaged projection EM overview of colon crypt region from the same animal. Areas with virus are highlighted (green asterisks) along with a region of interest for imaging by ET (yellow box). Virus was found in close proximity to a blood vessel (BV) containing red blood cells (RBCs). (**D**) Tomographic slice of region of interest from panel C showing separate pools of free virus on two sides of an infected cell. Black arrows indicate budding virions.

The following source data is available for figure 1:

**Source data 1.** Source data for *Figure 1A*.

## Tissue clearing and IF imaging of lymphoid tissues

We next sought to obtain spatial information about T-cell localization from multiple tissue types in PBMC-NSG mice. Optical clearing techniques offer the opportunity to image large volumes of opaque tissues by IF without serial sectioning (*Richardson et al., 2015*). We used different methods for clarifying tissues: GALT and female reproductive tract (FRT) were cleared using CLARITY/PACT (*Treweek et al., 2015*), and spleen samples containing heme were cleared using CUBIC (*Tainaka et al., 2014*) (*Figure 2A*). Tissues from uninfected PBMC-NSG animals were immunostained and imaged by confocal microscopy using antibodies against human CD3 to evaluate T-cell reconstitution and against HIV-1 p24 to rule out background staining in uninfected animals. IF of cleared GALT from uninfected animals revealed human CD3+ T-cells in the colon, spleen, and FRT (*Figure 2B*). These results established that optically cleared, intact lymphoid tissues from PBMC-NSG mice could be imaged by IF to detect the presence and localization of human T-cells.

Lymphoid tissue samples from an HIV-infected animal that had a high blood p24 level at 21 days PI (10.9 ng/mL p24) were cleared and stained to visualize the distribution of human CD3+ T-cells and HIV-1. Confocal microscopy in different lymphoid tissues from HIV-1–infected PBMC-NSG mice revealed the presence of human CD3+ T-cells, a subset of which were positive for HIV-1 p24 (*Figure 2C*). Some regions of p24 staining were not positive for human CD3, representing either free HIV-1 virions or infected cells other than T cells (*Figure 2C*).

## Longitudinal quantification of GALT infectivity

PBMC-NSG mice were infected and necropsied at specific times during the first six weeks PI (*Figure 3A*), and lymphoid tissues were prepared for imaging studies as described above. Cleared colon samples from HIV-1–infected or mock-infected controls were immunostained and imaged (*Figure 3B*). Representative images revealed low levels of infected human CD3+ T-cells at 5 days PI, which rapidly increased to peak levels at 10–16 days PI, before diminishing at time points >21 days PI. To enable quantification of T-cell and infection densities, the total number of nuclei in confocal slices were determined using the Fiji software suite (*Schindelin et al., 2012*), resulting in an agreement of 96.2 ± 1.9% compared to manual quantification (*Figure 3—figure supplement 1*). Human CD3+ T-cells from at least four confocal slices in 1–3 animals at individual time points were manually quantified in infected and uninfected controls (n > 5000 cells per time point). T-cells in colon from uninfected PBMC-NSG mice remained near 6% of total cells (*Figure 3C*), comparable to the ~10–12% found in uninfected human patient samples (*McElrath et al., 2013*). Human CD3+ T-cell densities in HIV-1–infected animals were similar to densities in uninfected controls until 12 days PI, at which time T-cell levels rapidly dropped to <4% of total cells at 16 days PI and continued to decrease gradually out to 45 days PI. p24+/human CD3+ T-cells were quantified similarly in GALT, revealing peak infectivity at 12 days PI, with nearly 3% of cells representing HIV-1–infected T-cells (*Figure 3D*). Infectivity levels rapidly decreased to <1% of cells representing infected T-cells after 21 days PI and approached zero by 35 days PI. When compared to p24 levels in blood from the same animals, the percent of p24+ T-cells peaked at 12 days PI and preceded peak blood p24 levels by 9 days, when p24 levels were low (*Figure 3D*). At peak blood p24 levels (21 days PI), the percentage of p24+ T-cells in colon had already diminished to nearly half of maximum levels at 10–12 days PI. These results are consistent with a disconnect between tissue infectivity and blood viral load over time, as previously noted in human samples (*Anton et al., 2003*; *Chun et al., 1997*). p24 not co-localized with human CD3+ T-cells was compared to levels of p24 associated with human CD3+ T-cells (*Figure 3B,E*). This analysis revealed an increase in p24 signal not associated with human CD3+ T-cells concomitant with decreased infectivity of human CD3+ T-cells at 21 days PI.

## Spatial and temporal distributions of HIV-1 in GALT

Intact regions of HIV-1-infected GALT cleared and immunostained for human CD4+ T-cells, human CD8+ T-cells, and HIV-1 p24 at specific times PI were imaged by confocal microscopy, and categories of cells in individual Z-slices were quantified (*Figure 4*). A 45 µm confocal Z-stack from the colon of an HIV-1–infected hu-mouse at 12 days PI displayed human CD4+ T-cells and p24+ cells localized to intestinal crypts and the bottom half of villi, whereas human CD8+ T-cells were dispersed uniformly throughout the crypts and the length of the villi (*Figure 4A,B* and *Video 1*). Mean distances from the base of crypts were 51.2 ± 34.5 µm for human CD4+ T-cells, 79.2 ± 36.8 µm for human

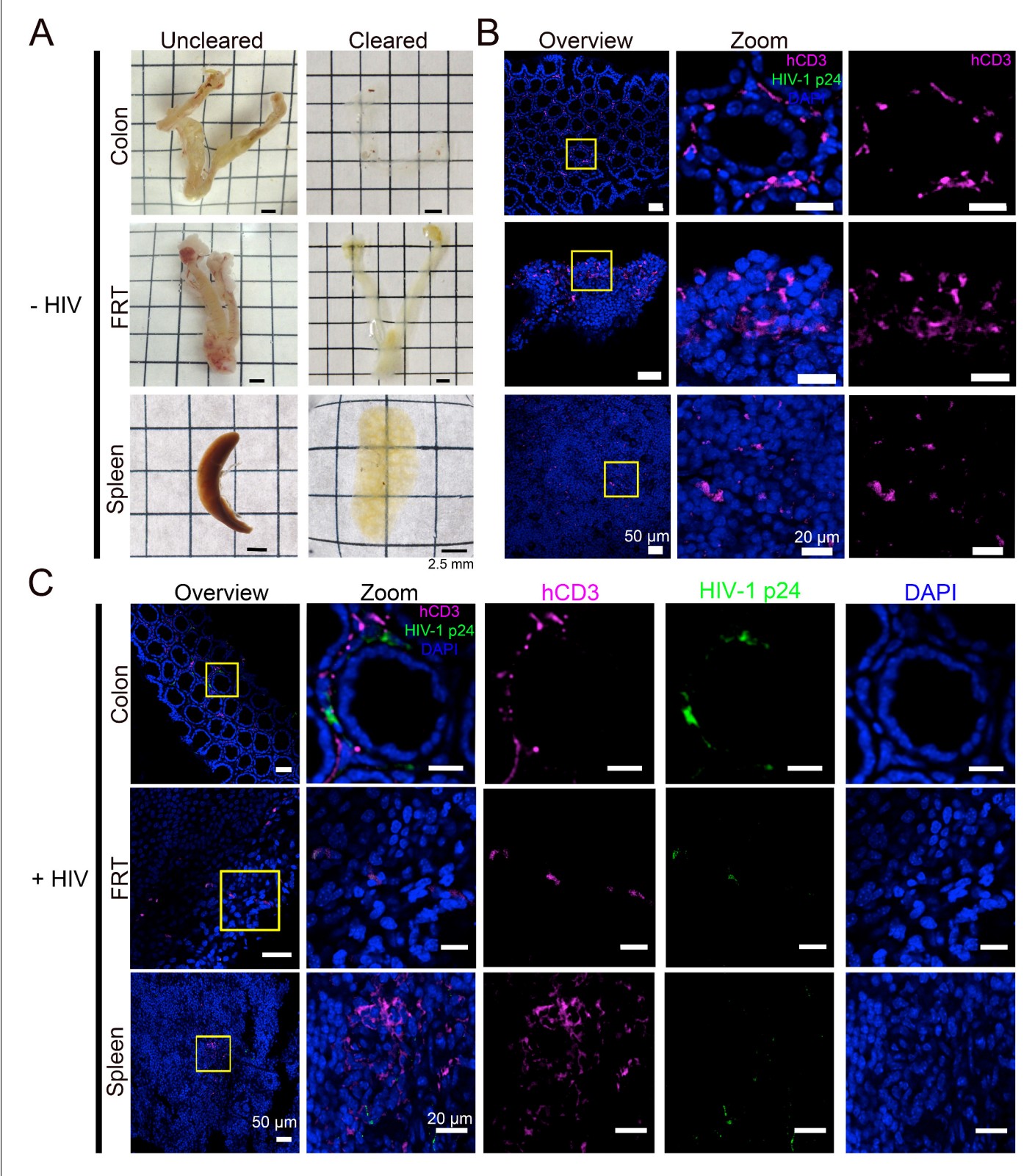

**Figure 2.** Tissue clearing and IF imaging of lymphoid tissues. (**A**) Representative images of uncleared (left) and cleared (right) lymphoid tissues from uninfected hu-mice. (**B, C**) Confocal images of cleared lymphoid tissues from uninfected (panel B) and HIV-1–infected (28 days PI) (panel C) hu-mice stained with antibodies against human CD3 (magenta) and HIV-1 p24 (green). Nuclei were labeled with DAPI (blue). Scale bars for overview images = 50 μm. Scale bars for zoomed views = 20 μm.

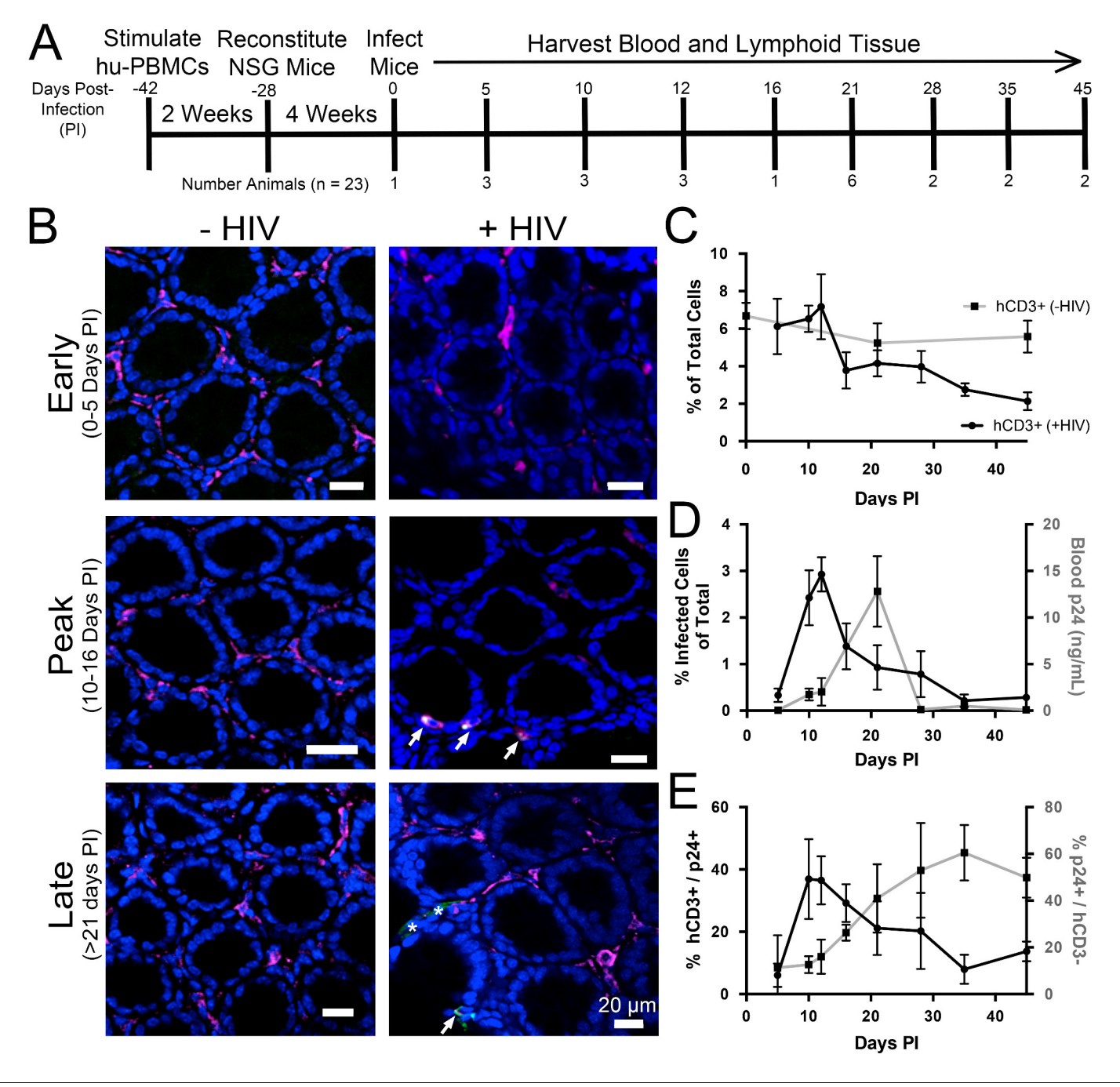

**Figure 3.** Longitudinal quantification of GALT infectivity. (A) Timeline of immune cell reconstitution, HIV-1 infection, and tissue harvest from hu-mice. Top numbers depict days PI, bottom numbers show number of animals per timepoint. Single uninfected animals were sacrificed at 0, 21, and 45 days PI. All other animals were infected and sacrificed at the times indicated. (B) Representative images showing human CD3+ T-cells (magenta), HIV-1 p24 (green), and nuclei (blue) in colon at early (0–5-days PI), peak (10–16 days PI), and late (>21 days PI) infectivity. Arrows indicate infected cells. Asterisks show p24 not associated with human CD3+ T-cells. (C–E) Error bars represent standard deviation. n > 5000 cells from 1–3 animals for all time points. (C) Quantification of human CD3+ T-cells as the percent of total cells for HIV-1–infected (black line) or uninfected (gray line) animals. (D) Comparison of GALT infectivity as percent infected cells of total (black line, left Y-axis) with blood p24 (gray line, right Y-axis, ng/mL) levels over time. (E) Longitudinal profile of percent HIV-1 p24-positive human CD3+ T-cells (black line, left Y-axis) and percent of p24 signal not associated with human CD3+ T-cells (gray line, right Y-axis).

The following source data and figure supplement are available for figure 3:

*Figure 3 continued on next page*

*Figure 3 continued*

**Source data 1.** Source data for *Figure 3C*.
**Source data 2.** Source data for *Figure 3D*.
**Source data 3.** Source data for *Figure 3E*.
**Figure supplement 1.** Validation of automated quantification of nuclei.

CD8+ T-cells, and 47.9 ± 45.1 µm for p24+ cells (*Figure 4C*). The localization of HIV-1–infected cells to colon crypts is consistent with our previous EM and IF studies of later stages of HIV-1 infection in GALT from BLT hu-mice (*Ladinsky et al., 2014*) and with results from early SIV infection in rhesus macaques (*Hirao et al., 2014*). The spatial arrangement of these cell types remained constant during the course of the experiment (*Figure 4D*), even as the total numbers of human CD4+ T-cells and p24+ cells markedly decreased at late time points (*Figure 4E*), likely due to target cell depletion.

## ET of longitudinal HIV-1 spread in GALT

To gain ultrastructural information about systemic virus spread in GALT, sections of HIV-1–infected colon were imaged by EM at representative time points (*Figure 5A*). Tissue samples adjacent to those used for IF were imaged in parallel by ET, as opposed to directly imaging the same tissue sample using both IF and EM, because the methods for extracting lipids from cleared tissues for IF eliminates scaffolding within the cell, resulting in destruction of ultrastructural details at the EM level (*Treweek et al., 2015*). EM surveys of colon crypts, where we previously characterized active HIV-1 infection by ET in a BLT hu-mouse model (*Ladinsky et al., 2014*), confirmed the infectious profile depicted by IF (*Figure 5A and B*). Early after infection (5 days PI), infected cells and free virus were not readily detected. This was followed by an exponential increase in the number of infected cells and the presence of large extracellular pools of free virus, often containing >100 free virions within 200–400 nm thick regions of tissue during peak infectivity (10–16 days PI). At this time, ~10% of crypts showed signs of active infection cells as evidenced by cells with budding virions attached to the plasma membrane (*Figure 5B*; center panel). At later times (>21 days PI), the total number of infected cells was reduced and individual pools of free virus in similar locations contained fewer virions. Most of the pools of free virus were not associated with actively-infected cells, consistent with our IF observation that levels of HIV-1 p24 not associated with human CD3+ T-cells increased at later time points PI (*Figure 3B and E*). These results suggested that most of the p24+/CD3- cells detected by IF at late times PI represented pools of free-virions, rather than infected cells other than T cells.

A cross section of a colon villus highlighting a vascularized central channel of lamina propria from an HIV-1–infected hu-mouse 10 days PI revealed large pools of virus (*Figure 5C*; left and center panels). Pools of virus in lamina propria were often found near vasculature, and ET allowed tracing of a virus-containing lateral intercellular space in 3D exhibiting pools of virus with openings leading to the walls of blood vessels (*Figure 5C*; right panels and *Video 2*).

## Parallel IF and ET of HIV-1 infection in spleen

We next imaged HIV-1 in spleen, first verifying that this organ in PBMC-NSG hu-mice exhibited characteristic architectural features. Splenic tissue can be histologically divided into regions of red and white pulp (*Mebius and Kraal, 2005*). Red pulp makes up a majority of the volume of the spleen, contains large numbers of red blood cells (RBCs) generating a red hue, and primarily functions to filter the blood. White pulp contains large numbers of B- and T-cells that coordinate immune responses within the blood. As such, spleen harbors large quantities of HIV-1 target cells, making it an important organ for HIV-1 pathology (*Mebius and Kraal, 2005*; *Ganusov and De Boer, 2007*; *Langeveld et al., 2006*). Indeed, a longitudinal study of SIV infection in NHP spleen revealed large numbers of HIV-1–infected CD4+ T-cells during acute infection that resulted in T-cell depletion and morphological changes in tissue at later stages of infection (*Williams et al., 2016*), consistent with human autopsy studies (*Diaz et al., 2002*).

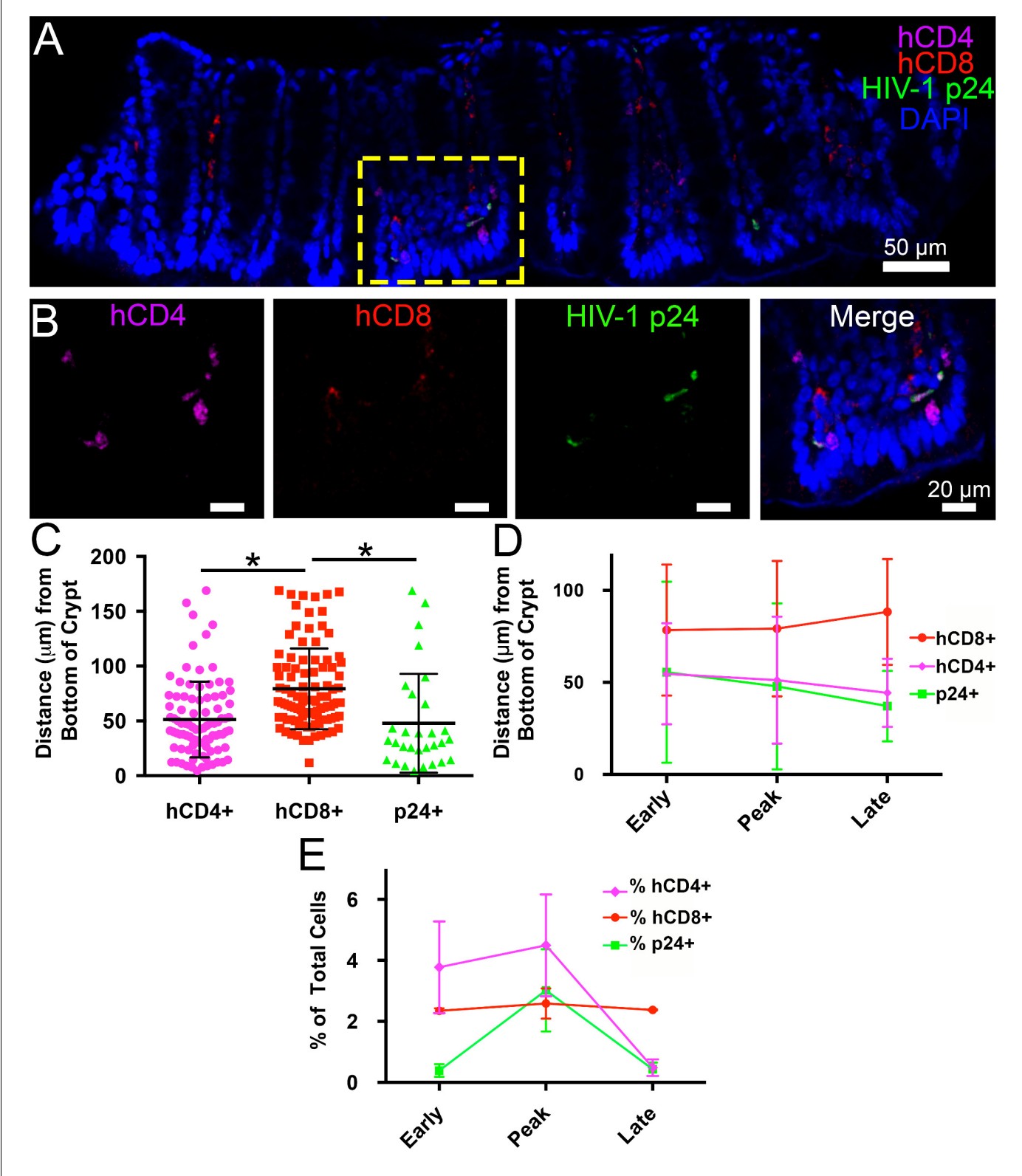

**Figure 4.** Spatial distribution of HIV-1 infection in GALT over time. (**A**) Representative confocal slice from 45 μm Z-stack depicting the distribution of human CD4+ T-cells (magenta), HIV-1 p24 (green), human CD8+ T-cells (red) and nuclei (blue) in colon at 12-days PI. (**B**) Zoom of boxed region in panel A showing individual channels and merged view for human CD4+ T-cells, human CD8+ T-cells, and p24. (**C–E**) Error bars represent standard deviation. (**C**) Quantification of Z-stack from panel A showing the distance (μm) from the base of crypts of human CD4+ T-cells (magenta), human CD8+ T-cells

*Figure 4 continued on next page*

*Figure 4 continued*

(red), and p24+ T-cells (green) for the volume in panel A. * indicates p<0.0001 (unpaired *t*-test with Welch's correction, two-tailed). (D) Quantification of distance (μm) from the base of crypts for human CD4+ T-cells (magenta), human CD8+ T-cells (red), and p24+ T-cells (green) for individual volumes during times of early, peak, and late infectivity. n > 2000 total cells for each volume. (E) Quantification of CD4+ T-cells (magenta), human CD8+ T-cells (red), and p24+ T-cells (green) as a percentage of total cells at early, peak, and late infectivity times. n > 5000 total cells for each time point.

The following source data is available for figure 4:

**Source data 1.** Source data for *Figure 4C*.
**Source data 2.** Source data for *Figure 4D*.
**Source data 3.** Source data for *Figure 4E*.

To determine if splenic tissue from hu-mice could be used to investigate HIV-1 infection, we imaged spleens from wildtype, NSG, and PBMC-NSG mice by brightfield, LM and EM to determine levels of human T-cell reconstitution and compare splenic architectures (*Figure 6* and *Figure 6—figure supplement 1*). Bright-field images of wildtype spleen showed the presence of round foci of white pulp separated by regions of red pulp (*Figure 6—figure supplement 1*). EM of white pulp from wildtype animals revealed diverse cell types, including lymphocytes, and collagen deposits. In contrast, brightfied microscopy of spleen from non-reconstituted NSG animals showed little to no white pulp, and EM of these samples indicated different cell types, denser packing, and little collagen (*Figure 6—figure supplement 1*). Brightfield microscopy images of uninfected PBMC-NSG animals revealed regions of white pulp at greater levels than non-reconstituted NSG animals, but these were not organized into foci as in wildtype spleen. IF of uninfected PBMC-NSG spleen showed localized pockets of human CD3+ T-cells interspersed throughout volumes of splenic tissue (*Figures 2B* and *6A*; left panels). Montaged projection EM surveys of uninfected PBMC-NSG spleen demonstrated delineations between white pulp and red pulp regions (*Figure 6B*; left panels and *Figure 6—figure supplement 1*). Red pulp contained enucleated RBCs, while white pulp was distinguished by an absence of RBCs, bundles of collagen fibrils, and populations of lymphocytes identified as cells with large, dense nuclei surrounded by minimal cytoplasm. Overall, the organization of the spleen in PBMC-NSG hu-mice was intermediary between non-reconstituted NSG animals and wildtype immunocompetent animals, with white pulp occurring as smaller striations within the red pulp as opposed to the circular foci seen in spleens from wildtype animals (*Figure 6—figure supplement 1*). These results demonstrated that structural and cellular components of splenic white pulp, most importantly HIV-1 target cells, are present in PBMC-NSG animals even though the splenic architecture is altered.

To evaluate the distribution of virus and T-cells, adjacent samples of spleen from an HIV-1–infected PBMC-NSG animal at 12 days PI were imaged by IF and EM. IF of spleen revealed the presence of human CD3+ T-cells in addition to sporadic foci of HIV-1 p24 distributed throughout large regions of tissue (*Figure 6A*, right panels). EM of spleen from the same animal showed discernable regions of white pulp containing

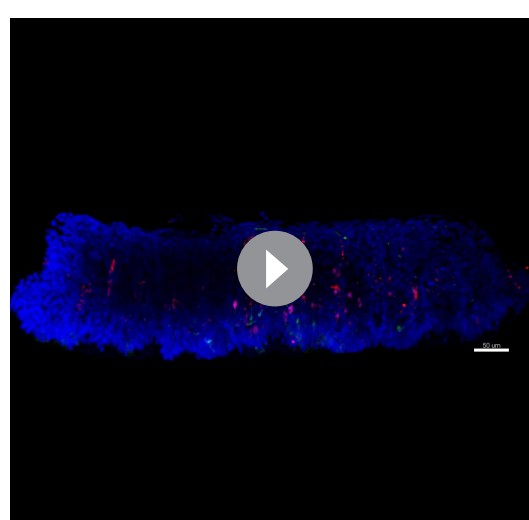

**Video 1.** Spatial Distribution of HIV-1 and T-cells in GALT 360˚ rotation of a 45 μm thick Z-stack depicting the spatial distribution of human CD4+ T-cells (magenta), HIV-1 p24 (green), human CD8+ T-cells (red) and nuclei (blue) in colon at 12-days PI. Crypts are at the base and villus tips extend toward the top of the volume. Quantification of this volume is reported in *Figure 4A–C*.

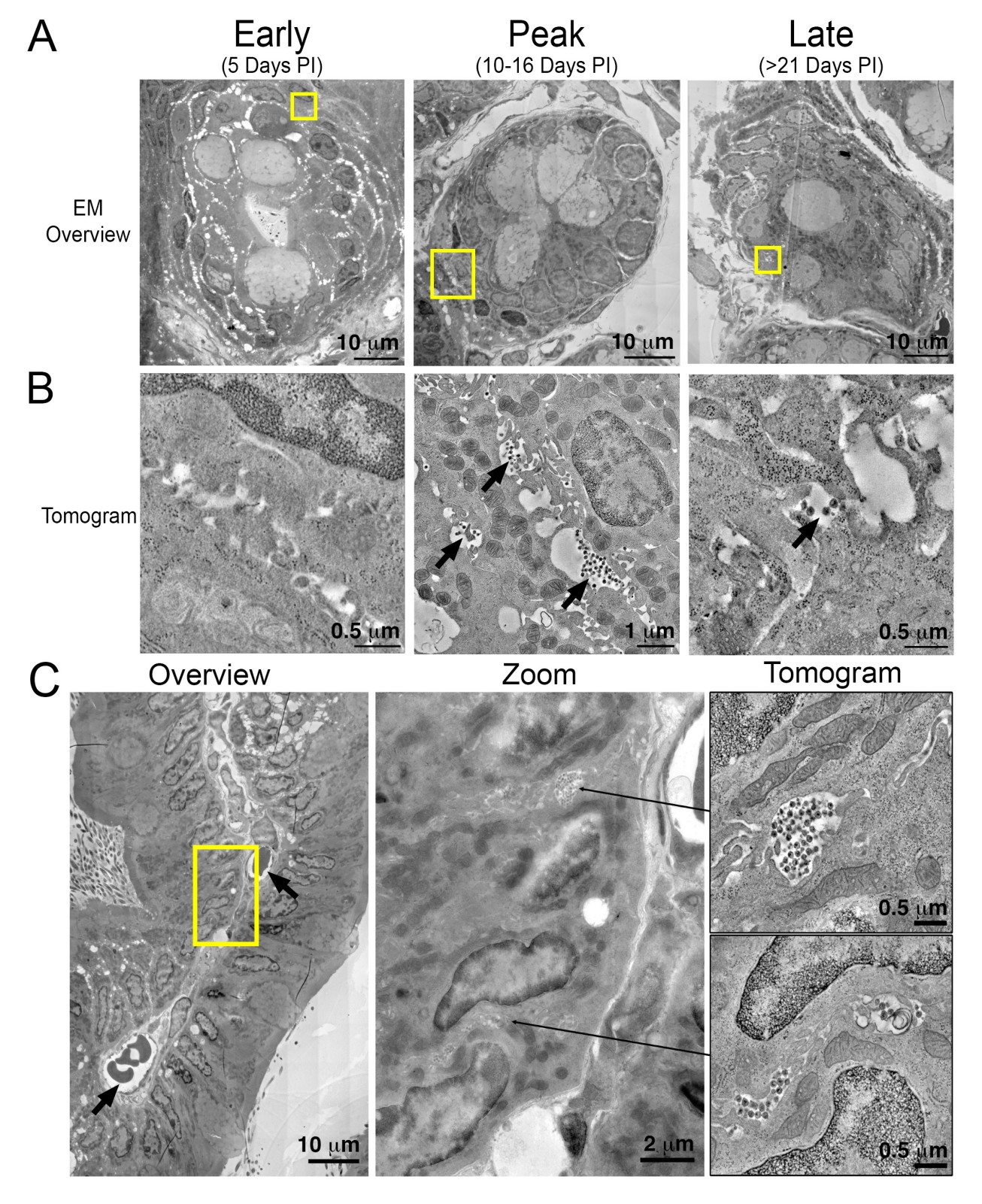

**Figure 5.** Electron tomography of longitudinal HIV-1 spread in GALT. (**A, B**) Montaged overview EM projection images (panel A) and zoomed tomographic slices (panel B) of HIV-1–infected colon crypts from hu-mice from early (5 days), peak (10–16 days), and late (>21 days) time points PI. Arrows indicate pools of free virions between cells. (**C**) Lamina propria from HIV-1 infected GALT 10 days PI. Left: montaged overview EM projection image showing lamina propria in cross-section. Arrows indicate blood vessels containing red blood cells. Yellow box indicates zoomed region of

*Figure 5 continued on next page*

*Figure 5 continued*

interest depicted in middle panel. Zoomed region from overview reveals pools of virus adjacent to vasculature. Right: tomographic slices of two indicated regions (arrows) that reveal pools of virus contiguous with blood vessel walls connected by lateral intercellular space.

infected lymphocytes intercalated within bundles of collagen fibrils and surrounded by HIV-1 virions (*Figure 6B*; right panel and *Figure 7A,B*; top panels). ET of an HIV-1–infected cell from spleen 12 days PI revealed prolific virus production at distinct regions of plasma membrane (*Figure 7A*). Serial section tomographic reconstruction of a 6.9 µm x 3.5 µm x 0.8 µm (19.3 µm$^3$) volume from a single infected cell contained nearly 1000 virus particles of which 846 were mature, 18 were immature, and 102 were budding from the cell surface (*Figure 7A* and *Video 3*). By approximating the volume of an entire cell as an ellipsoid, we used this result to calculate that a single cell could produce >6000 virions during a single round of virus production. A reconstructed 3.4 µm x 3.4 µm x 1.2 µm (13.9 µm$^3$) volume from another HIV-1–infected cell in spleen at the same time PI revealed 1350 virus particles with 1329 mature, 15 immature, six budding virions, and predicted >4000 virions released (*Figure 7B*).

## HIV-1 budding machinery

VPS4 is a hexameric AAA+ ATPase (*Monroe et al., 2014*) recruited by the endosomal sorting complex required for transport (ESCRT) protein network to enable the fission of membranes during cellular processes including cell division, endosomal sorting, and virus budding (*Sundquist and Kräusslich, 2012*). In previous ET studies of HIV-1 infection in GALT from BLT hu-mice, we identified discreet densities localized at the base of budding virions and proposed that they included VPS4 oligomers (*Ladinsky et al., 2014*). We found similar densities localizing to the base of budding virions in tomograms of splenic tissue from HIV-1-infected PBMC-NSG mice at 12 days PI (*Figure 8A*). Immuno-EM using an anti-VPS4 antibody showed localization of VPS4 to the base of budding virions (*Figure 8B*), demonstrating that VPS4 localizes to the same region, suggesting that the densities include VPS4, perhaps in complex with other proteins.

In some cells, ET revealed large numbers of virions (>100 budding profiles) produced from distinct regions of the plasma membrane of HIV-1–infected cells (*Figure 8C*). In the cytoplasm immediately adjacent to these sites of prolific virus release, we found accumulations of large numbers of densities indistinguishable from the densities near budding virions (*Figures 8C,D* and *9*). These cytoplasmic densities were found in direct proximity (within 1 µm of the plasma membrane) to large numbers of budding virions and contained hundreds to thousands of densities within an individual accumulation. As the cytoplasmic accumulations of these densities were rare, we were unable to prepare analogous negatively-stained samples for immuno-EM. However, we ruled out identification of the densities as ribosomes, since they were more electron dense than neighboring ribosomes and also lacked the characteristic cleft between the small and large ribosomal subunits (*Figure 9*).

The cytoplasmic densities were not found in infected lymphocytes in the splenic white pulp that were surrounded by large pools of free virions but displayed few budding profiles (<10 buds); large cytoplasmic pools of density were absent in a cell that exhibited only six budding profiles (some of which also included the

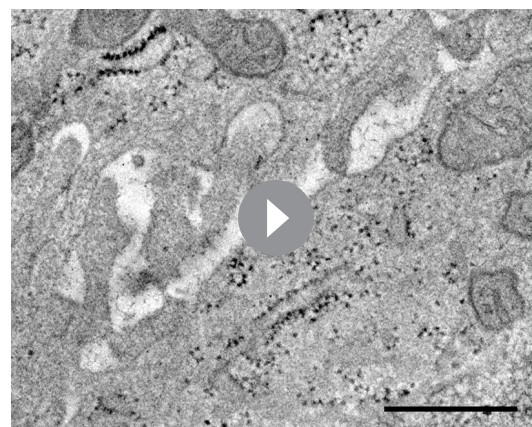

**Video 2.** ET of a pool of virions in GALT at peak infectivity Tomographic reconstruction of a volume of two consecutive 400 nm serial sections from colon of a PBMC-NSG animal 10-days PI (*Figure 5C*, upper right panel). A pool of virions resides in the center with the lateral intercellular space between two cells extending into the upper right portion of the movie and is contiguous with a blood vessel wall (*Figure 5C*, center panel). Scale bar = 0.5 µm.

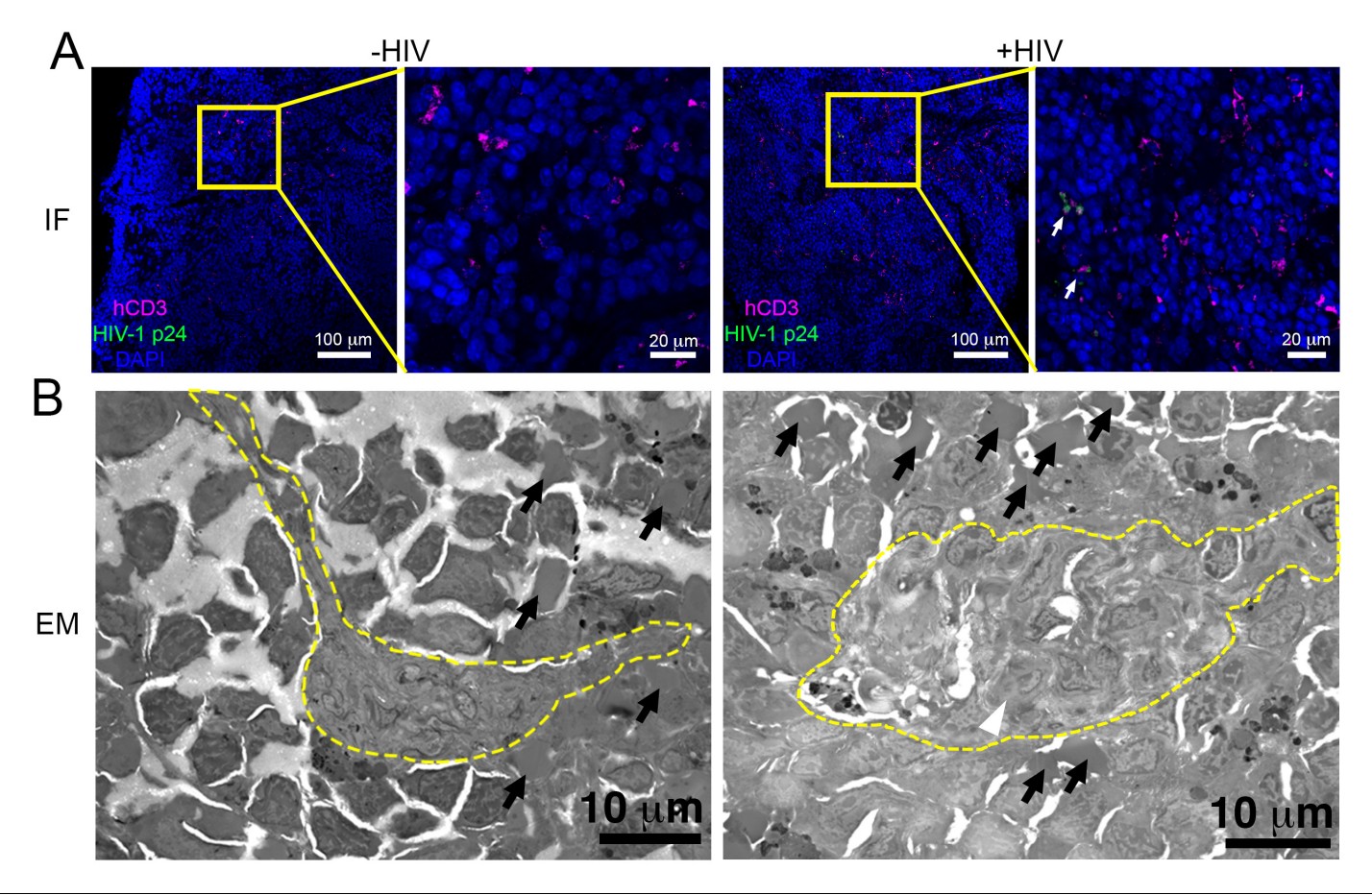

**Figure 6.** Parallel IF and ET of HIV-1 infection in spleen. (**A**) Confocal IF images of cleared spleen showing human CD3+ T-cells (magenta), HIV-1 p24 (green), and nuclei (blue) from an uninfected (-HIV) or infected (+HIV) animal at 12-days PI. Yellow boxes indicate areas of zoomed views. White arrows indicate regions with p24. (**B**) Montaged overview EM projection images from adjacent regions of spleen from the same animals in A. Yellow dashed lines depict regions of white pulp, black arrows indicate red blood cells in a region of red pulp, white arrowhead shows an HIV-1-infected cell.

The following figure supplement is available for figure 6:

**Figure supplement 1.** Architecture of the spleen in wildtype, NSG, and PBMC-NSG mice.

densities at the bases of the buds) but was surrounded by >1000 free extracellular virions (*Figure 8D*). Indeed, accumulations of cytoplasmic densities were only seen in cells undergoing prolific virus budding, and in these cells, only in regions adjacent to the buds (*Figure 8D* and *Video 4*). These results suggest that cytoplasmic pools of these densities accumulate during periods of prolific virus production and are exhausted following a round of virus release.

## Discussion

Here we combined tissue clearing, 3D-IF microscopy, and ET to longitudinally assess T-cell systemic spread of HIV-1 in a hu-mouse model of HIV-1 infection at multiple levels of volume and resolution. Tissue clearing and IF microscopy allowed interrogation of volumes of intact tissues at single cell resolution. This approach provided information about the spatial distribution of human immune cells, HIV-1–infected cells, and virus proteins within multiple tissues from PBMC-NSG hu-mice at specific times after infection. ET of adjacent tissue regions allowed imaging at subcellular resolution and validated infectivity patterns identified by IF. These studies revealed a rapid and vast spread of virus throughout GALT and spleen at times when virus in blood was barely detectable.

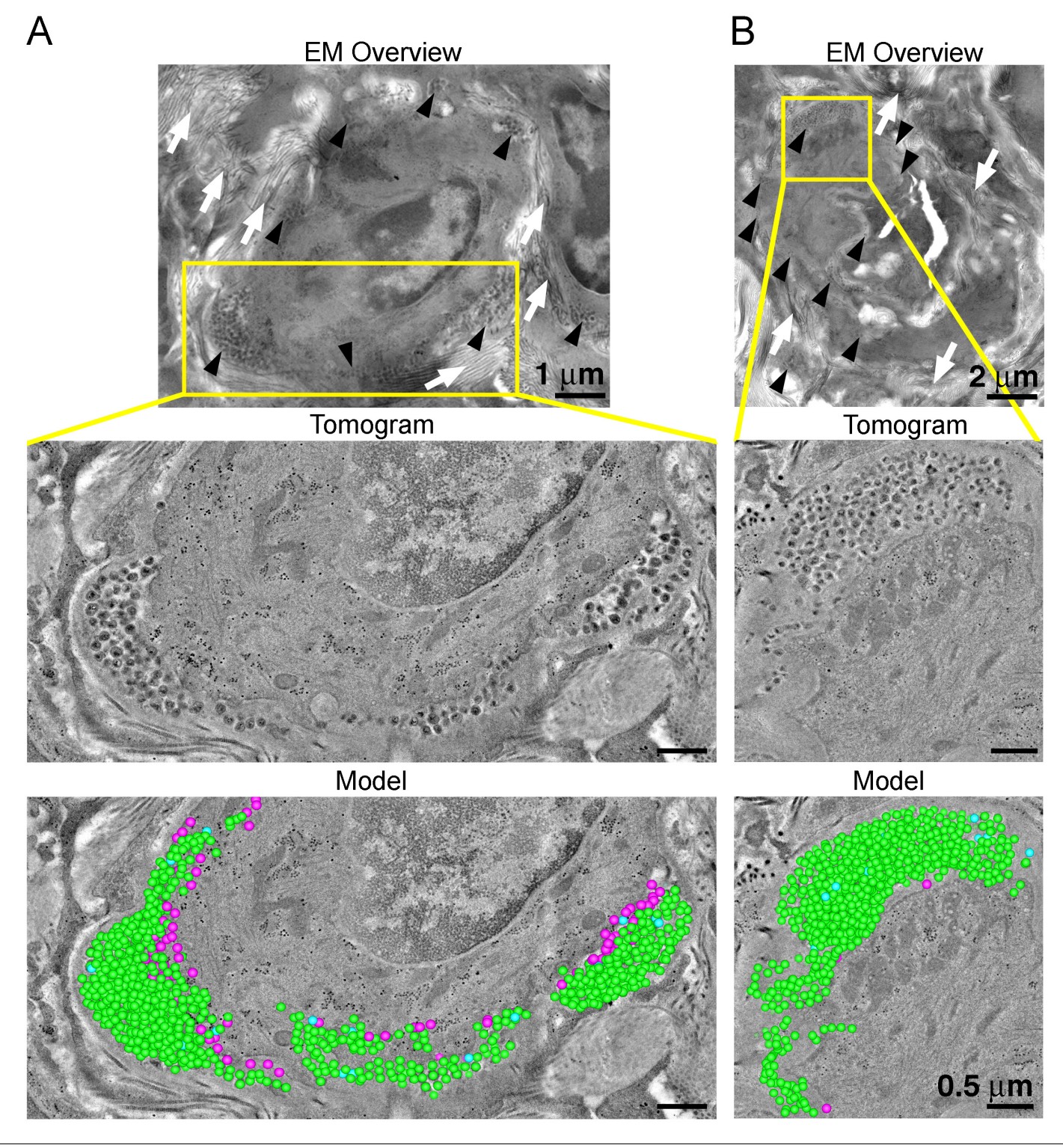

**Figure 7.** ET and modeling of HIV-1 Infection in spleen. (**A** and **B**) EM projection images of HIV-1-infected cells from regions of spleen from an HIV-1-infected animal 12-days PI (top panels). Collagen fibrils (white arrows) and pools of virions (black arrows) surround the infected cells. Yellow boxes indicate regions of tomographic reconstruction. Tomographic slices show zoomed detail of infected cells (middle panels). Modeled mature (green), immature (blue), and budding (magenta) virions from the full volume of tomographic reconstructions (bottom panels). Scale bars for tomogram and model panels are 0.5 m.

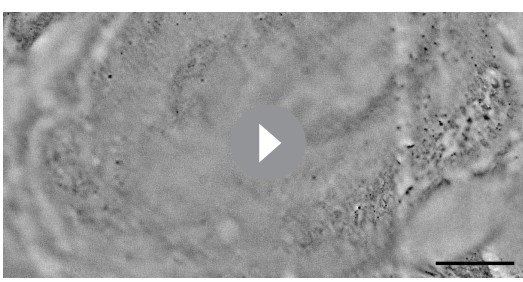

**Video 3.** Modeling of an infected cell and individual virions in spleen Tomographic reconstruction of a volume of two consecutive 400 nm serial sections of an HIV-1–infected cell from spleen of a PBMC-NSG animal 12-days PI. Virions are modeled as mature (green), immature (blue), and budding (magenta). Coiled fibrils of collagen are evident surrounding the periphery of the cell and free virions. The second half of the movie shows zoomed detail. Scale bar = 1 μm.

One focus of these studies was GALT, an important site for early aspects of HIV-1 interaction with the immune system that is home to a large proportion of HIV-1 target cells and is critical for early establishment of HIV-1 infection and pathogenic effects that lead to long-term disease progression (*Guy-Grand and Vassalli, 1993*; *Saksena et al., 2010*; *Cerf-Bensussan and Guy-Grand, 1991*; *Belmonte et al., 2007*; *Mestecky et al., 2009*). Tissue clearing and 3D-IF allowed the quantification of human T-cells and HIV-1–infected cells within intact GALT from PBMC-NSG hu-mice (*Figure 3*) and revealed a human CD3+ T-cell density that was comparable to those from human patient colon samples (*McElrath et al., 2013*). Furthermore, human T-cells were found in intestinal crypts, in agreement with results from HIV-1–infected colon samples from BLT hu-mice (*Ladinsky et al., 2014*), colon samples from SIV-infected NHP (*Mohan et al., 2007*), and uninfected human colon samples (*McElrath et al., 2013*), validating the PBMC-NSG hu-mouse model for studying HIV-1 infection of human T-cells in GALT. Longitudinal quantification of infectivity in GALT revealed the dynamics of HIV-1 infectivity over time. Peak infectivity in GALT occurred while levels of virus in the blood, a hallmark of HIV-1 progression, were quite low. GALT infectivity then rapidly decreased and coincided with a drop in human CD3+ T-cell density in HIV-1–infected animals (*Figure 3C,D*), consistent with studies in humans showing T-cell depletion in GALT during acute HIV-1 infection (*Brenchley et al., 2004*; *Mehandru et al., 2004*). In our previous ET study of HIV-1-infection in GALT from BLT animals, all samples were from 10–20 weeks PI, yet HIV-1 and infected T cells were present (*Ladinsky et al., 2014*). The more rapid dynamics of virus spread and target cell depletion in GALT of HIV-1–infected PBMC-NSG hu-mice could be due to the high levels of T cell activation and a lack of T cell replenishment in PBMC-NSG animals (*Marsden and Zack, 2015*).

In the colon of HIV-1–infected animals, human CD4+ T-cells preferentially localized to the bottom third of villi and in crypts whereas human CD8+ T-cells were more ubiquitously distributed (*Figure 4* and *Video 1*). Accordingly, HIV-1–infected human CD4+ T-cells also localized to the bottom third of villi and in crypt regions, with most infected cells residing in crypts. This spatial distribution remained consistent, even as human CD4+ andCD8+ T-cell levels changed during the course of infection (*Figure 4D,E*). These results suggest that local tissue environments influence cell localization and distribution within hu-mice; comparisons between other animal models and human patient samples will be required to extrapolate these findings.

Adjacent sections of HIV-1–infected GALT were prepared for analysis by ET to gain ultrastructural information that focused on the establishment and spread of HIV-1 in lymphoid tissues. ET of GALT from HIV-1–infected PBMC-NSG hu-mice revealed few or no infected cells or virions at 5 days PI, peak infectivity at 10–16 days PI when infected cells and large pools of free virions were readily detectable, and reduced numbers of infected cells and sizes of virus pools at >21 days PI (*Figure 5*). The ET results were consistent with our IF results and also agree with studies showing the establishment of SIV-infection in GALT and spleen early after infection (*Hirao et al., 2014*; *Williams et al., 2016*).

Hundreds to thousands of free virions accumulating in pools near infected cells in both GALT and spleen were detected by ET (*Figure 5, 7* and *Videos 2* and *3*), consistent with ET of GALT from HIV-1–infected BLT hu-mice (*Ladinsky et al., 2014*). The highest levels of free virus in both GALT and spleen occurred at 10–12 days PI, a time when virus levels in the blood were low (*Figure 3*). We detected pools of free virus contiguous with mucosal vasculature in GALT (*Figure 5*), and in spleen, a tissue that acts to filter the blood, we visualized >1000 virions being produced from one region of an HIV-1–infected cell (*Figure 7*). We calculated that a single infected cell within tissue could

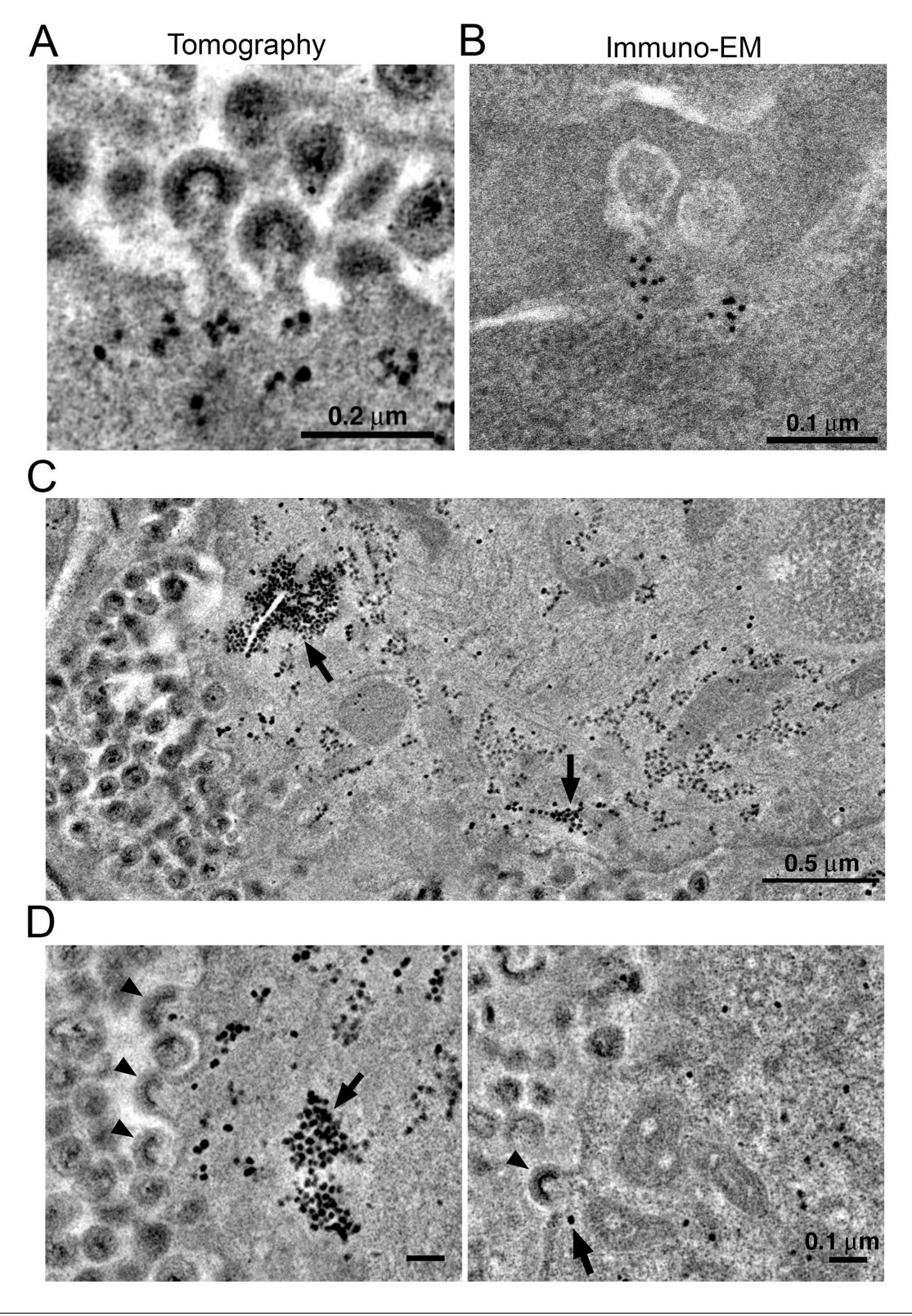

**Figure 8.** Localization of HIV-1 budding machinery. (**A**) Densities localized to the base of budding virions. (**B**) Anti-VPS4A immuno-EM revealing localization patterns consistent with densities in panel A. (**C**) Tomographic slice of HIV-1–infected spleen 12 days PI showing large accumulations of densities (black arrows) adjacent to regions of prolific virus release. (**D**) Tomographic slices showing budding virions from the animal in panel C revealing prolific budding (left) or low levels of budding (right). Arrowheads indicate budding virions and arrows indicate densities.

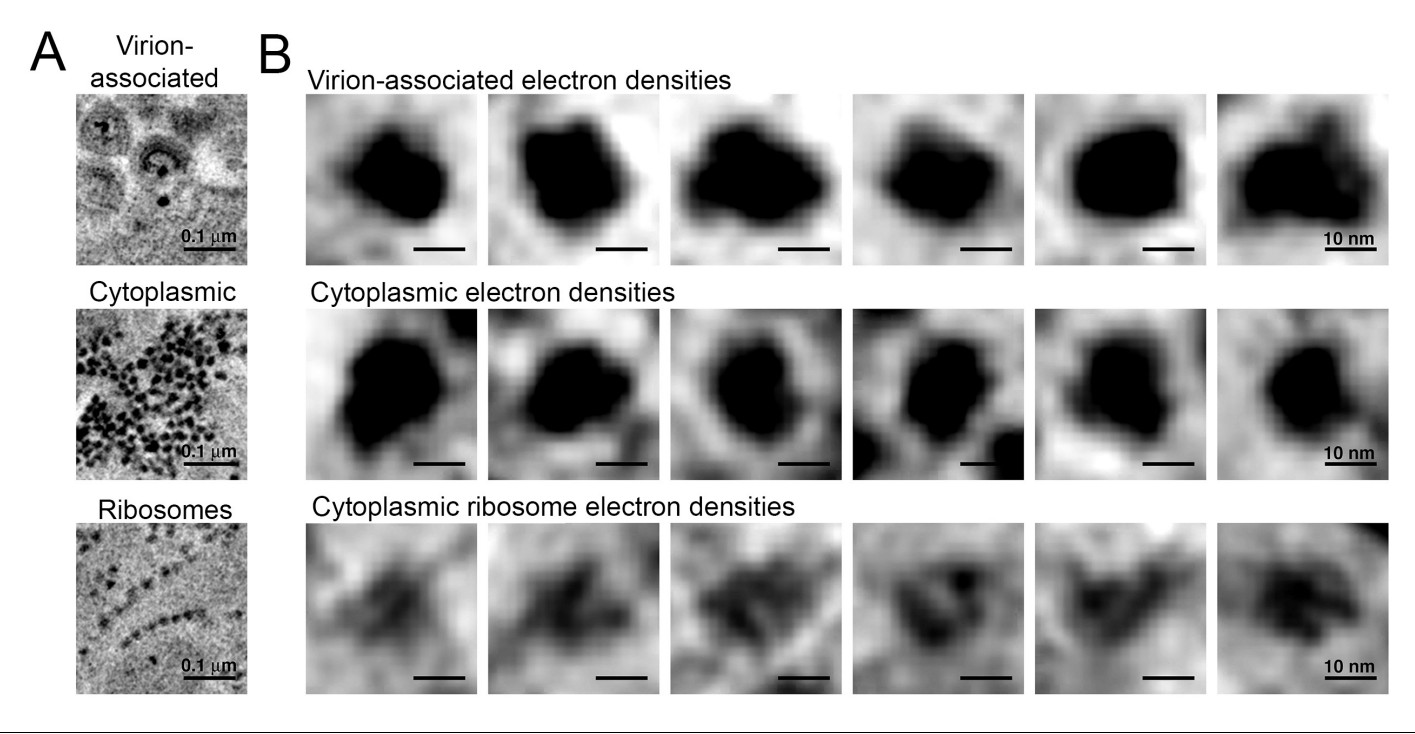

**Figure 9.** Comparison of virion-associated densities, cytoplasmic densities, and densities corresponding to cytoplasmic ribosomes. (**A**) Representative tomographic slices of densities associated with budding virions (top), cytoplasmic pools near budding virions (middle), and cytoplasmic ribosomes. (**B**) Gallery of images with the same tomographic thickness (9.1 nm) for individual densities associated with budding virions (top), cytoplasmic pools near budding virions (middle), and cytoplasmic ribosomes. Dimensions of virion-associated and cytoplasmic densities were indistinguishable. Ribosomes contain a visible cleft between individual subunits that is absent in the densities associated with budding virions and cytoplasmic pools near areas of prolific virus production.

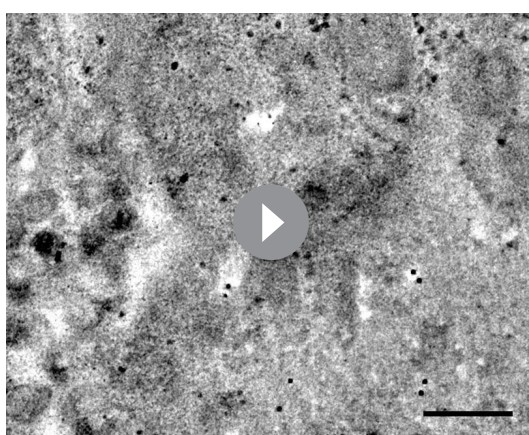

**Video 4.** ET of densities associated with virus budding Tomographic reconstruction of an accumulation of cytoplasmic densities associated with prolific virus release. Densities were present in cytoplasm immediately adjacent to a region of plasma membrane supporting prolific virus release and at the base of budding virions. Scale bar = 0.2 μm.

produce >6000 virions during a round of virus release, providing insights into the exponential production of HIV-1 within tissues early after infection. The total number of SIV virions produced by an individual infected cell was previously estimated to be $4.0–5.4 \times 10^4$ (*Chen et al., 2007*). If a similar number of HIV-1 virions are produced from infected cells within tissue, this implies multiple rounds of virus release from each infected cell, as suggested by our previous ET studies (*Ladinsky et al., 2014*). In lymphoid tissues, the large numbers of target cells combined with high levels of virus production and accessibility to vasculature suggest that HIV-1–infected cells in tissue provide a source of free virus that contributes to blood viral load and systemic spread.

We previously proposed that distinct densities observed at the base of budding virions during virus release contained VPS4 (*Ladinsky et al., 2014*), an AAA+ ATPase recruited by polymerized ESCRT-III proteins that acts to recycle ESCRT-III in conjunction with

abscission of the nascent virus membrane (*Sundquist and Kräusslich, 2012*). Here we showed immuno-EM localization of anti-VPS4A antibodies to equivalent regions of budding virions (*Figure 8*). In cells undergoing prolific budding, we found aggregations of hundreds to thousands of similar densities in adjacent regions of cytoplasm within close proximity (<1 μm) to budding virions (*Figure 8* and *Video 4*). Although we could not conclusively identify these densities as containing VPS4, they differed in size, intensity and shape from ribosomes, which are also distributed throughout the cytoplasm (*Figures 8* and *9*). Notably, the large accumulations of cytoplasmic densities were only present in cells undergoing prolific budding (>100 buds), suggesting they are directly involved in HIV-1 assembly and/or budding. Future immuno-EM studies will be required to directly identify their components and involvement in the HIV-1 replication cycle.

In summary, the parallel multiscale imaging approaches applied here allowed visualization of early HIV-1 spread in a hu-mouse model of infection at both single cell and subcellular resolution. 3D-IF of optically-cleared tissue allowed us to longitudinally and spatially assess the distribution of human T-cells and virus proteins within volumes of intact lymphoid tissue. ET of adjacent regions of tissue allowed the detection of individual virions, infected cells, and subcellular protein complexes within the context of actively-infected tissues at specific times PI. These approaches are directly applicable to additional lymphoid tissues such as lymph nodes, bone marrow, and thymus in infected hu-mouse, NHP, and human patient samples. Future efforts will allow the visualization of discreet biological processes occurring during early HIV-1 spread in lymphoid tissues and enable the development of a quantifiable spatial map of HIV-1 spread in tissues during early infection.

## Materials and methods

### Antibodies

Primary antibodies for flow cytometry: mouse monoclonal anti-human CD3-FITC (1:13; Biolegend), mouse monoclonal anti-human CD4-PE (1:13; Biolegend), mouse monoclonal anti-human CD8α-APC (1:13; Biolegend). Primary antibodies for IF: rabbit polyclonal anti-human CD3 (1:200; Dako), mouse monoclonal anti-HIV-1 p24 (1:200; Dako), rabbit polyclonal anti-HIV-1 p24 (1:200; gift of Wesley Sundquist, University of Utah), mouse monoclonal anti-human CD4-AlexaFluor 488 (1:50; Invitrogen), mouse monoclonal anti-CD8-PE (1:100; Biolegend). Secondary antibodies for IF: Alexa Fluor 488 goat anti-rabbit IgG, Alexa Fluor 546 goat anti-rabbit IgG, Alexa Fluor 647 goat anti-rabbit IgG, Alexa Fluor 488 goat anti-mouse IgG, Alexa Fluor 546 goat anti-mouse IgG, Alexa Fluor 647 goat anti-mouse IgG, Alexa Fluor 488 donkey anti-goat IgG, Alexa Fluor 546 donkey anti-goat IgG, and Alexa Fluor 647 donkey anti-goat IgG (Invitrogen). All secondary antibodies were used at 1:1000. For immuno-EM, rabbit monoclonal anti-VPS4A (AbCam) was used at a dilution of 1:500 and 10 nm gold-conjugated goat anti-rabbit IgG (Ted Pella) was used at a dilution of 1:20.

### Flow cytometry of primary and reconstituted human PBMCs

Cultured human PBMCs (AllCells, inc.) were analyzed by flow cytometry (MACSquant) to verify the presence of T-cell populations as described (*Balazs et al., 2012*) using antibodies against human CD3, CD4, and CD8. Lymphocyte populations were identified by forward and side-scatter profiles and human T-cells were identified by human CD3 staining. Gated human CD3+ T-cell subsets were evaluated for the presence of human CD4 and CD8 T-cells. Mouse whole blood samples were treated with RBC lysis/fixation solution (Biolegend) prior to immunostaining and analysis by flow cytometry as described for cultured huPBMCs.

### Humanization and HIV-1 infection of immune deficient mice

NOD.Cg-*Prkd*^scid^*Il2rg*^tm1Wjl^/SzJ (NOD/SCID/IL2Rγ−/− or NSG); The Jackson Laboratory) mice were reconstituted with human PBMCs as previously described (*Kumar et al., 2008*; *Balazs et al., 2012*; *Nakata et al., 2005*). Briefly, primary human PBMCs (AllCells, inc.) were cultured for 13 days in RPMI simultaneously supplemented with human IL-2 (R and D Biosystems) and continually stimulated with phytohemagglutinin-P (Sigma). 2–5 × 10^6 cells were injected intraperitoneally into each animal. Reconstituted animals were retro-orbitally infected with 25 ng R5-tropic HIV-1 (pNL4-3 backbone pseudotyped with a YU2 envelope obtained from Michel Nussenzweig, Rockefeller University) four weeks later. Infection was monitored by weekly blood draw and upon sacrifice. Blood samples were

inactivated with 4% Triton X100 for 1 hr at 37°C, and p24 levels were quantified by ELISA (Cell Biol-abs). Assuming 2000 molecules of p24 per virion, and a molecular weight of 24 kDa, there are ~$10^7$ HIV virions per nanogram of p24 (http://tronolab.epfl.ch/webdav/site/tronolab/shared/protocols/TUvsp24.html). Animals were sacrificed by $CO_2$ inhalation followed by cervical dislocation and nec-ropsied. Lymphoid tissues were excised and immediately placed into cacodylate buffer containing 5% sucrose (Sigma) and 0.1 M sodium cacodylate trihydrate (Sigma) with freshly added 8% parafor-maldehyde (Electron Microscopy Sciences). Samples were stored in fixative at 4°C until processed for LM, EM, or immuno-EM. All animal experiments were approved by the Caltech IACUC.

## Tissue clearing

GALT and FRT samples were cleared using the CLARITY/PACT method adapted from (*Treweek et al., 2015*). Fixed tissue regions were washed 3x in PBS for 30 min each and incubated at 4°C overnight in a hydrogel monomer solution of 10% acrylamide in PBS supplemented with 0.25% photoinitiator VA-044 (2,20 -Azobis[2-(2-imidazolin-2-yl) propane] dihydrochloride; Fischer). Tissue samples were warmed to room temperature, incubated for 1–3 hr at 37°C, and monitored for hydrogel polymerization as indicated by a rapid increase in solution viscosity. Samples were immedi-ately placed into PBS and washed 3x over the course of several hours. Tissue samples in hydrogel were placed into conical tubes containing 8% SDS in PBS and incubated for 2–7 days at 37°C with shaking. SDS was exchanged daily until samples appeared visually transparent. Cleared tissue hydro-gel samples were washed 5x in PBS over one day and stored in the dark in PBS at room temperature for up to 6 months.

Spleen sample were cleared using the CUBIC method adapted from *Susaki et al. (2014)*. Fixed tissue samples were washed in PBS as described above, placed into CUBIC-1 solution (25% w/v urea, 25 w/v *N,N,N',N'*-tetrakis (2-hydroxypropyl) ethylenediamine (Sigma), 15% w/v polyethylene glycol mono-*p*-isooctylphenyl ether/Triton X-100 (Sigma), and incubated at 37°C with gentle agita-tion. CUBIC-1 solution was exchanged daily for 3–7 days until tissue was adequately decolorized. Samples were then washed 5x in PBS over one day, transferred into CUBIC-2 solution (50% w/v sucrose, 25% w/v urea, 10% w/v 2,2,2'-nitrilotriethanol (Sigma), and 0.1% (v/v) Triton X-100), and incubated at 37°C with gentle agitation. CUBIC-2 solution was exchanged daily for 5–10 days until the tissue appeared visually transparent. CUBIC-2–treated tissue samples were washed 5x in PBS over one day and either immediately immunostained, stored for up to 6 months in PBS at room tem-perature in the dark, or frozen and stored long term in optimal cutting temperature compound (Sakura Finetek) at −80°C.

## Immunostaining

Cleared tissue samples were incubated in blocking buffer (2–4% FBS, 0.1% Tween-20 or Triton X-100, 0.01% $NaN_3$) overnight at 4°C. For samples that were immunostained with mouse monoclonal antibodies, rat anti-mouse CD16/32 (Biolegend) was added to blocking buffer (1:50) to reduce back-ground. Primary antibody staining was conducted in blocking buffer containing primary antibodies for 2–3 days. Tissue samples were washed 5x over one day with wash solution (0.1% Tween-20 or Tri-ton X-100, 0.01% $NaN_3$ in PBS). Samples were then incubated with secondary antibody (1:1000) in blocking buffer for 2 days and washed 5x over one day in wash solution. To stain for nuclei, washed tissue samples were incubated in 1 μg/mL 4',6-diamidino-2-phenylindole (DAPI) in PBS for 8 min. Samples were washed 3x for 10 min in PBS and incubated overnight in refractive index matching solution (RIMS) containing 90% Histodenz (Sigma) in 0.02M $NaPO_4$ (Sigma) pH 7.5 buffer supple-mented with 0.1% Tween-20 and 0.01% $NaN_3$ (Sigma) for CLARITY/PACT-cleared tissues or CUBIC-2 solution for CUBIC-cleared tissues. All staining and incubations were performed at room tempera-ture with gentle rocking and kept in the dark after the addition of fluorescent-labeled antibodies.

## Confocal IF microscopy and image analysis

Samples were mounted between two No.1 coverslips (Electron Microscopy Sciences) separated by adhesive/adhesive silicone isolators (Electron Microscopy Sciences) in RIMS for CLARITY samples or CUBIC-2 solution for CUBIC samples. Mounted samples were imaged with a Zeiss LSM-710 or 880 using a LD LCI Plan-Apochromat 25 × 0.8 NA Imm Corr DIC M27 multi-immersion objective (w.d. 0.57 mm) with glycerol. Images were analyzed with the Fiji software suite (*Schindelin et al., 2012*).

Images were smoothed and individually thresholded to minimize background tissue autofluorescence. Nuclei were quantified by thresholding the DAPI channel to show individual nuclei. Images were smoothed, made binary, and particles >2 µm were analyzed after overlapping nuclei were separated using the watershed function. Automated nuclei quantification was compared against four manual counts with >650 individual nuclei per field of view. 3D movies of confocal Z-stacks were made with the Imaris software suite. Gamma levels for DAPI staining in movies only were thresholded to emphasize CD4, CD8, and p24 cell staining throughout the volume.

### Brightfield imaging

Fixed spleen tissues were also used for brightfield imaging. Large slices (~2 mm x 3 mm by x 0.5 mm) of tissue were cut with a microsurgical scalpel and placed on glass slides. Specimens were imaged with a Diaphot-200 microscope (Nikon, Inc.) using a 4x objective lens in brightfield mode. Images were recorded digitally.

### EM sample preparation

Prefixed lymphoid tissues (described above) were prepared for EM as previously described (*Ladinsky et al., 2014*). Briefly, samples were rinsed in cacodylate buffer and trimmed to <~1 mm$^3$ blocks. One to three tissue blocks were placed into brass freezing planchettes (Ted Pella, Inc.) pre-filled with cacodylate buffer containing 10% Ficoll (Sigma). Samples were rapidly frozen in a HPM-010 high pressure freezer (Leica Microsystems) and transferred to liquid nitrogen. The planchettes were placed under liquid nitrogen into cryo tubes (Nunc) containing 2 ml of 2.5% OsO$_4$, 0.05% uranyl acetate in acetone, and then transferred to a AFS2 freeze-substitution machine (Leica Microsystems). Samples were freeze substituted for 72 hr at −90°C and then warmed to −20°C over 12 hr. After 24 hr at −20°C, the samples were warmed to room temperature, rinsed with acetone, and infiltrated with Epon-Araldite resin (Electron Microscopy Sciences). Tissue pieces were flat embedded between two Teflon-coated glass slides and polymerized at 60°C for 48 hr.

Embedded tissue samples were selected and sectioned as previously described (*Ladinsky et al., 2014*). 400 nm sections were cut with a UC6 ultramicrotome (Leica Microsystems) using a diamond knife (Diatome, Ltd.) and placed on Formvar-coated copper-rhodium 1 mm slot grids (Electron Microscopy Sciences). Sections were stained with 3% uranyl acetate and lead citrate. Colloidal gold particles (10 nm) were placed on both surfaces of the grids as fiducial markers. Grids were stabilized with evaporated carbon prior to imaging.

### Electron tomography

Grids were placed in a Dual-Axis Tomography holder (Fischione Instruments, Inc.) and imaged with a Tecnai TF30ST-FEG microscope (FEI) at 300 KeV. Dual-axis tilt series (±64°; 1° intervals) were acquired automatically using SerialEM software (*Mastronarde, 2008*). Tomographic data were aligned, analyzed and segmented using IMOD (*Mastronarde, 2005*) on MacPro computers.

### Immuno-EM

Prefixed lymphoid tissues were cut into ~1 mm$^3$ blocks and infiltrated with 2.1 M sucrose in cacodylate buffer over 10 hr. Blocks were affixed to aluminum sectioning stubs and frozen in liquid nitrogen. Cryosections (100 nm) were cut with a UC6/FC6 cryoultramicrotome using a Cryo-Immuno diamond knife (Diatome, Ltd), transferred in a wire loop containing a drop of 2.3 M sucrose in PBS to Formvar-coated, carbon-coated, glow- discharged 100-mesh copper/rhodium grids, warmed to room temperature, labeled with primary and colloidal gold secondary antibodies, negatively stained, and embedded with 1% uranyl acetate, 1% methylcellulose in dH$_2$O. Stained grids were air-dried in wire loops overnight.

## Acknowledgements

We thank Andres Collazo and the Caltech Biological Imaging Center for use of the Zeiss LSM-710 and 880 confocal microscopes and help with image capture and analysis, Carol Garland and Alasdair McDowall for help maintaining electron microscopes, Viviana Gradinaru and Ben Deverman for advice about tissue clearing methodologies, Wesley Sundquist for a rabbit polyclonal anti-HIV-1 p24

antibody, and Michel Nussenzweig for the pNL4-3envYU2 HIV-1. This work was supported by the National Institutes of Health (2 P50 GM082545-06; WI Sundquist, PI), Ragon Institute and Rosalind W. Alcott post-doctoral fellowships (CK), gifts from the Gordon and Betty Moore Foundation and the Agouron Institute to support electron microscopy at Caltech, and funds provided by The Regents of the University of California, Research Grants Program Office, California HIV/AIDS Research Program, Grant Number ID15-CT-017. The opinions, findings, and conclusions herein are those of the author and not necessarily represent those The Regents of the University of California, or any of its programs.

## Additional information

### Competing interests
PJB: Reviewing editor, *eLife*. The other authors declare that no competing interests exist.

### Funding

| Funder | Grant reference number | Author |
| --- | --- | --- |
| National Institute of General Medical Sciences | 2 P50 GM082545-06 | Pamela J Bjorkman |
| California HIV/AIDS Research Program | ID15-CT-017 | Pamela J Bjorkman |

The funders had no role in study design, data collection and interpretation, or the decision to submit the work for publication.

### Author contributions
CK, Conceptualization, Formal analysis, Methodology, Writing—original draft, Writing—review and editing; MSL, Data curation, Formal analysis, Investigation, Writing—original draft, Writing—review and editing; AN, Data curation, Methodology, Writing—review and editing; RPG, Conceptualization, Data curation, Formal analysis, Methodology, Writing—review and editing; PJB, Conceptualization, Formal analysis, Supervision, Funding acquisition, Writing—original draft, Writing—review and editing

### Author ORCIDs
Pamela J Bjorkman, http://orcid.org/0000-0002-2277-3990

### Ethics
Animal experimentation: This study was performed according to the recommendations in the Guide for the Care and Use of Laboratory Animals of the National Institutes of Health. All of the animals were handled according to approved institutional animal care and use committee (IACUC) protocols (#1702) of the California Institute of Technology. Full effort was made to minimize suffering during animal sacrifice.

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
