## [Decision Letter]

Thank you for submitting your article "Longitudinal imaging of HIV-1 spread in humanized mice with parallel 3D immunofluorescence and electron tomography" for consideration by *eLife*. Your article has been favorably evaluated by Jeremy Luban (Reviewing Editor) and Wenhui Li (Senior Editor), and four reviewers. The following individuals involved in review of your submission have agreed to reveal their identity: Jeremy Luban (Reviewer #1) and Lishan Su (Reviewer #3).

The reviewers have discussed the reviews with one another and the Reviewing Editor has drafted this decision to help you prepare a revised submission.

Summary:

This manuscript presents a set of descriptive experiments that define exciting and technically complex methodology for assessment of acute HIV-1 infection in an in vivo small animal model. What is particularly exciting here is the ability of the authors to follow the virus at multiple levels of scale. While the significance of findings from such a model remains to be determined it has potential to be extremely useful for studying early events of HIV-1 infection in vivo that will be difficult to address using other experimental systems.

There was significant disagreement among the four reviewers about the significance of this manuscript. All reviewers agree that the mouse model system used here is not "ideal" and that it suffers from significant limitations that confound interpretation of the work. In the end, we have decided that it would be reasonable to go forward with publication, given that the manuscript is modified to clearly explain the limitations of the model.

Essential revisions:

1) The authors introduce the Hu-PBMC model with an emphasis on its utility without acknowledging the potential limitations of the system relative to other humanized mouse models or other animal models. They should note up front that the Hu-PBMC model is a T cell dominant, xenoreactive model where the T cells are all activated, and most other lineages are not represented. The relatively high level of T cell activation of all T cells in this system could lead to high levels of virus production from infected cells in these mice. Overall, the manuscript makes technical advances and acquires images that are novel, but it remains to be seen what one may observe when examining acutely infected human tissues.

2) PHA blasts-NSG mice can be infected by HIV. However, it is not the most relevant model to study early HIV-1 infection and spread in lymphoid tissues. In this study, human PBMCs were stimulated with PHA/IL-2 for two weeks before adoptive transfer into NSG mice. HIV-1 infection/spread in mice transferred with these mostly activated T cells should be very different from HIV infection of naïve hosts with multiple cell types. The limitation of the PHA blasts-NSG model needs to be discussed. Key findings should be confirmed with BLT or other humanized mice with a quiescent human immune system with multiple immune cells in multiple lymphoid organs.

3) In Figure 3, the authors show that nearly 50% of infected cell in GALT were hCD3- cells. It is of interest to identify/discuss what such hCD3- p24+ cells are.

4) Figure 3. The authors should also represent the total CD4 T cell depletion so that we can see to what extent to which the decrease in infected cells at late time points is simply due to target cell depletion.

5) This study is described as a "parallel" study because "adjacent GALT samples from the colon were fixed with aldehydes for parallel IF and ET analysis". A stronger effort to correlate the data between the light and electron microscopy would be necessary to allow them to be more conclusive in their observations of non-CD3 associated virus clusters in ET.

6) One reviewer made the following comment. Perhaps the authors have some additional data that would move the manuscript in the kind of direction that they recommend. The authors do not delve deeply enough into using their imagining techniques to add new knowledge to the field of HIV mucosal transmission and pathogenesis. Using these techniques to compare routes of infection (intravaginal vs. intrarectal vs. intravenous) could be highly informative, as well as testing how various viral isolates, perhaps those with different chemokine receptor tropisms, may disseminate differently throughout the GALT. Even within the data currently presented, additional follow-up could be conducted to fully interrogate the results, such as the imaging of p24+ cells that were interestingly not co-localized with CD3+ T cells. The characterization of these infected cell types or determining whether these images represent cell-free virus warrants significant follow-up experiments.

7) Again, the authors do not adequately discuss the caveats of their model, which may have significant impact on the interpretation of the results by the reader. Though it has been shown that CD4+ T cells are indeed rapidly depleted from gut tissues during acute HIV infection, the depletion seen here is quite rapid and extensive (virtually permanent) and may be an important caveat of the model. In contrast to natural infection, here new T cells are not being generated from human-derived hematopoietic stem cells in the bone marrow, as would be the case in the BLT or Hu-HSC humanized mouse models. This important caveat needs to be discussed in more detail to give an accurate interpretation of the results, particularly with respect to the time course data.

---

## [Author Response]

Essential revisions:

1) The authors introduce the Hu-PBMC model with an emphasis on its utility without acknowledging the potential limitations of the system relative to other humanized mouse models or other animal models. They should note up front that the Hu-PBMC model is a T cell dominant, xenoreactive model where the T cells are all activated, and most other lineages are not represented. The relatively high level of T cell activation of all T cells in this system could lead to high levels of virus production from infected cells in these mice. Overall, the manuscript makes technical advances and acquires images that are novel, but it remains to be seen what one may observe when examining acutely infected human tissues.

We thank the reviewers for pointing out that the paper would be improved by more explicitly addressing limitations of the PBMC-NSG humanized mouse model. The revised paper includes a longer discussion of this mouse model, including the important caveats raised by the reviewers.

We are eager to examine acutely-infected human tissues using the imaging methods described here – the problem with doing this has been that we have been unable to obtain human samples that have been preserved in a way that allows high resolution electron tomography. In addition, even if we had human samples, it is not possible to survey multiple types of infected tissues in humans or to perform equivalent longitudinal studies.

We were also unable to conduct longitudinal imaging experiments in BLT mice comparable to ones we reported in the submitted paper (which required 23 total mice) because of the prohibitive expense, labor, and time required to generate these mice and the lack of expertise at Caltech in making BLT mice. Our choice of PBMC-NSG mice as the humanized mouse model for our studies was influenced by previous research done at Caltech in David Baltimore’s laboratory. Because my laboratory learned how to make and infect hu-mice from the Baltimore lab, we used their mouse model for our studies: specifically, we used the PBMC-NSG hu-mouse model described in the Baltimore laboratory paper demonstrating that AAV delivery of anti-HIV antibodies is protective against HIV infection [1]. In this paper, Balazs et al. showed data demonstrating that the PBMC-NSG hu-mouse model exhibits CD4 T cell depletion upon challenge with HIV, and they noted that this had also been shown by others [2]; thus, we were confident that this mouse model could be used to reproduce the T cell-mediated systemic spread and subsequent T cell depletion that occurs in natural HIV infections. The Baltimore laboratory also used the PBMC-NSG hu-mouse model for some of the experiments described in their other papers [3,4]. In addition, a recent paper characterizing and describing the PBMC-NSG hu-mouse model demonstrated T cell depletion, high virus p24 levels in blood, and the effectiveness of both HAART and passive antibody transfer upon HIV infection in this mouse model, concluding that the model is a relevant infection and pathogenesis model [5].

2) PHA blasts-NSG mice can be infected by HIV. However, it is not the most relevant model to study early HIV-1 infection and spread in lymphoid tissues. In this study, human PBMCs were stimulated with PHA/IL-2 for two weeks before adoptive transfer into NSG mice. HIV-1 infection/spread in mice transferred with these mostly activated T cells should be very different from HIV infection of naïve hosts with multiple cell types. The limitation of the PHA blasts-NSG model needs to be discussed.

Please see the first paragraph of the Results section in the revised manuscript for a discussion of the limitations of the hu-mouse model used for our studies. To directly address the concerns of the reviewers, we point out that nearly all of the CD3+ T cells in the PBMC-NSG hu-mouse model are activated as compared to ~10% in healthy human patients [6]. The revised manuscript now notes the higher levels of activated T cells in PBMC-NSG mouse model compared with humans could alter the dynamics of HIV spread as compared to other hu-mouse models or human patients with lower levels of activated T cells and a more diverse set of immune cell lineages. However, because (i) activated T cells make up ~10% of the total T cell population in humans, (ii) T cell activation increases to 20-40% during the course of HIV infection [6], and (iii) T cells represent the largest population of HIV target cells in the body [7], our demonstration by ET that individual activated T cells can produce thousands of free virions provides a striking observation that is relevant for understanding the routes and modes of HIV-1 spread in lymphoid tissues.

Key findings should be confirmed with BLT or other humanized mice with a quiescent human immune system with multiple immune cells in multiple lymphoid organs.

We would like to conduct comparative imaging studies using tissues from other mouse models, but were unable to obtain BLT tissues other than GALT from our collaborators at the Ragon Institute who constructed the BLT mice we used for our previous imaging studies [8]. However, we have done direct comparisons of HIV-infected GALT from the PBMC-NSG mice used for the current studies and from the Ragon Institute BLT mice. We found that the localization and density of human T cells were equivalent in GALT from the two hu-mouse models (Figure 10). In addition, free virions were found at similar levels and locations in GALT from both mouse models (Figure 11). The observation that T cell levels in GALT were comparable between BLT and PBMC-NSG mice suggests that our results in PBMC-NSG mice represent a valid model to look at T cell-mediated spread of HIV. In addition, although we acknowledge that PBMC-NSG mice are reconstituted mainly with activated T cells, we again note that T cells make up a majority of all HIV target cells in humans [7] and T cell activation increases to 20-40% during the course of HIV infection [6]. Thus, the PBMC-NSG model represents a relevant model to study the spread of virus caused by activated T cells within lymphoid tissues.

Author response image 1.Equivalent IF confocal slices of colon crypts from HIV-1-infected BLT (left) or PBMC-NSG (right) hu-mice.The density and localization of human CD3+ T cells (green) are similar in both models (blue = DAPI nuclear stain).**DOI:**
http://dx.doi.org/10.7554/eLife.23282.024

Author response image 2.Equivalent tomographic slices of colon crypt regions from HIV-1-infected BLT (left) or PBMC-NSG (right) hu-mice.Free virions are visible at similar levels and locations in both models (red arrows = pools of free virions).**DOI:**
http://dx.doi.org/10.7554/eLife.23282.025

We’d also like to point that some aspects of the PBMC-NSG and other comparable hu-mouse models might be advantageous for our studies of early HIV spread. A recent review by Marsden and Zack [9] states, “In some respects HIV infection of the original hu-PBL-SCID and SCID-hu models emulate the early stages of HIV infection in humans, with exponential virus replication accompanied by profound CD4 depletion within weeks of infection in the absence of a specific antiviral immune response. In contrast, some of the newer hu-HSC and BLT models provide an environment that is more typical of later stages of HIV infection in humans, with specific primary cellular and (albeit limited) humoral immune responses present and slower CD4 declines."

*3) In Figure 3, the authors show that nearly 50% of infected cell in GALT were hCD3- cells. It is of interest to identify/discuss what such hCD3- p24+ cells are.*

The quantification of fluorescence in Figure 3 shows that ~50% of the p24+ fluorescence did not localize with hCD3 fluorescence at time points >28 days post-infection. The resolution of light microscopy cannot distinguish whether the p24+/hCD3- fluorescence represents non-T cells that were infected and actively producing virus, or if it represents pools of free virus that accumulated in lateral intercellular spaces surrounding cells. Parallel analysis by ET at late time points post-infection revealed pools of free virions surrounding cells (Figure 5). Thus, as clarified in the revised manuscript in the first paragraph of the "ET of longitudinal HIV-1 spread in GALT" portion of the Results section, we believe that much of the p24+/hCD3- fluorescence represents pools of free virions.

4) Figure 3. The authors should also represent the total CD4 T cell depletion so that we can see to what extent to which the decrease in infected cells at late time points is simply due to target cell depletion.

The results in Figure 4 directly address this point as human CD4+ T cell levels (magenta) drop at the late time point along total with p24+ cells (green), while CD8+ T cells (red) remain constant. We have revised the text in the "Spatial and temporal distributions of HIV-1 in GALT" portion of the Results section to emphasize that target cell depletion is likely occurring at late time points.

5) This study is described as a "parallel" study because "adjacent GALT samples from the colon were fixed with aldehydes for parallel IF and ET analysis". A stronger effort to correlate the data between the light and electron microscopy would be necessary to allow them to be more conclusive in their observations of non-CD3 associated virus clusters in ET.

Although methods for correlative fluorescent and EM imaging of the same sample have been developed (i.e., correlated light and electron microscopy or CLEM) [10], these technologies would not yield useful information in our studies because we used optically-cleared samples in order to image large volumes of tissue. The methods for extracting lipids from cleared tissues for IF eliminates scaffolding within the cell, resulting in destruction of ultrastructural detail at the EM level as evidenced by EM images of cleared samples in a recent paper we co-authored with Viviana Gradinaru’s laboratory [11] and a direct comparison shown here of the loss of ultrastructural information in a CLARITY sample (Figure 12). Thus, we cannot directly image the exact same tissue sample by both IF and ET. By parallel imaging, we mean imaging of pieces of adjacent tissues by IF and for EM, as clarified in the revised text in the first paragraph of the "ET of longitudinal HIV-1 spread in GALT" portion of the Results section.

Regarding the non-CD3-associated virus clusters, as explained above, ET suggests that these mainly represent pools of free virions.

Author response image 3.Equivalent tomographic slices of brush border regions of gut epithelial cells from wild-type mice that were optimally preserved for EM imaging by high-pressure freezing/freeze substitution (HPF/FS) fixation (left) or optically cleared using CLARITY prior to HPF/FS (right).Clearing caused drastic loss of ultrastructural detail, likely due to extraction of lipids from the tissue (Term. web = terminal web; ICS = intercellular space).**DOI:**
http://dx.doi.org/10.7554/eLife.23282.026

6) One reviewer made the following comment. Perhaps the authors have some additional data that would move the manuscript in the kind of direction that they recommend. The authors do not delve deeply enough into using their imagining techniques to add new knowledge to the field of HIV mucosal transmission and pathogenesis. Using these techniques to compare routes of infection (intravaginal vs. intrarectal vs. intravenous) could be highly informative, as well as testing how various viral isolates, perhaps those with different chemokine receptor tropisms, may disseminate differently throughout the GALT.

We would like to compare different routes of infection as the reviewer suggests, but the PBMC-NSG hu-mouse model is not suited for intravaginal or intrarectal routes of infection. For this reason, the Baltimore laboratory at Caltech used BLT mice for the majority of their studies of vaginally-infected hu-mice [3]. Because Caltech does not have a facility to produce BLT mice, the Baltimore laboratory obtained the mice from a collaborator, Dong Sung An, at UCLA. We have recently begun a collaboration with Dr. An to obtain longitudinal BLT tissue samples from mice infected by different routes with either CCR5- versus CXCR4-tropic HIV strains. We are also collaborating with other laboratories to obtain SIV- and SHIV-infected NHP tissue for imaging. However, these are on-going studies that will take at least a year to complete, so we cannot include them in the present manuscript.

Even within the data currently presented, additional follow-up could be conducted to fully interrogate the results, such as the imaging of p24+ cells that were interestingly not co-localized with CD3+ T cells. The characterization of these infected cell types or determining whether these images represent cell-free virus warrants significant follow-up experiments.

As described above, our ET results suggest that most of the CD3-/p24+ fluorescence resulted from pools of free virions rather than from CD3-/p24+ infected cells.

7) Again, the authors do not adequately discuss the caveats of their model, which may have significant impact on the interpretation of the results by the reader. Though it has been shown that CD4+ T cells are indeed rapidly depleted from gut tissues during acute HIV infection, the depletion seen here is quite rapid and extensive (virtually permanent) and may be an important caveat of the model. In contrast to natural infection, here new T cells are not being generated from human-derived hematopoietic stem cells in the bone marrow, as would be the case in the BLT or Hu-HSC humanized mouse models. This important caveat needs to be discussed in more detail to give an accurate interpretation of the results, particularly with respect to the time course data.

We agree these points are important to consider when interpreting our results and have now included a discussion of these caveats in the first paragraph of the Results section and the second paragraph of the Discussion.

References

1) Balazs, AB, Chen, J, Hong, CM, Rao, DS, Yang, L, Baltimore, D (2012) Antibody-based protection against HIV infection by vectored immunoprophylaxis. Nature 481, 81–84.

2) Kumar, P, Ban, HS, Kim, SS, Wu, H, Pearson, T, Greiner, DL, Laouar, A, Yao, J, Haridas, V, Habiro, K, Yang, YG, Jeong, JH, Lee, KY, Kim, YH, Kim, SW, Peipp, M, Fei, GH, Manjunath, N, Schultz, LD, Lee, SK, Shankar, P (2008) Cell-specific siRNA delivery suppresses HIV-1 infection in humanized mice. Cell 134, 577–586.

3) Balazs, AB, Ouyang, Y, Hong, CM, Chen, J, Nguyen, SM, Rao, DS, An, DS, Baltimore, D (2014) Vectored immunoprophylaxis protects humanized mice from mucosal HIV transmission. Nature Medicine, doi:10.1038/nm.3471.

4) Luo XM, Lei MYY, Feidi RA, West AP Jr, Balazs AB, Bjorkman, PJ, Yang, L Baltimore D (2010) Dimeric 2G12 as a Potent Protection against HIV-1. PLoS Pathog 6(12): e1001225.

5) Kim, KC, Choi, BS, Kim, KC, Park, KH, Lee, HJ, Cho, YK, Kim, SI, Dim, SS, Oh, YK, Kim, YB (2016) A Simple Mouse Model for the Study of Human Immunodeficiency Virus. AIDS Res Hum Retroviruses 32, 194-202.

6) Malingham, M, Peakman, M, Davies, ET, Pozniak, A, McManus, TJ, Vergani, D (1993) T cell activation and disease severity in HIV infection. Clin Exp Immunol. 93,337-43.

7) Haase AT (1999) Population biology of HIV-1 infection: viral and CD4+ T cell demographics and dynamics in lymphatic tissues. Annu Rev Immunol. 17:625-56.

8) Ladinsky, MS, Kieffer, C, Olson, G, Deruaz, M, Vrbanac, V, Tager, AM, Kwon, DS, Bjorkman, PJ (2014) Electron tomography of HIV-1 infection in gut-associated lymphoid tissue. PLoS Pathog 10: e003899.

9) Marsden and Zack, hu-mouse review, 2015

10) De Boer, P, Hoogenboom, JP, Giepman, BNG (2015) Correlated light and electron microscopy: ultrastructure lights up! Nature Methods 12, 503-513.

11) Treweek, JB, Chan, KY, Flytzanis, NC, Yang, B, Deverman, BE, Greenbaum, A, Lignell, A, Xiao, C, Cai, L, Ladinsky, MS, Bjorkman, PJ, Fowlkes, CC, Gradinaru, V (2015) Whole-body tissue stabilization and selective extractions via tissue-hydrogel hybrids for high-resolution intact circuit mapping and phenotyping. Nature Protocols 10: 1860-96.